



# Deep maxima of phytoplankton biomass, primary production and bacterial production in the Mediterranean Sea during late spring

Emilio Marañón[1], France Van Wambeke[2], Julia Uitz[3], Emmanuel S. Boss[4], María Pérez-Lorenzo[1], Julie Dinasquet[5], Nils Haëntjens[4], Céline Dimier[3], Vincent Taillandier[3]

[1]Department of Ecology and Animal Biology, Universidade de Vigo, 36310 Vigo, Spain

[2]Aix-Marseille Université, CNRS, Université de Toulon, CNRS, IRD, Mediterranean Institute of Oceanography, MIO UM 110, 13288 Marseille, France

[3]CNRS and Sorbonne Université, Laboratoire d'Océanographie de Villefranche, 06230 Villefranche-sur-mer, France

[4]School of Marine Sciences, University of Maine, Orono, Maine, USA

[5]Scripps Institution of Oceanography, University of California, San Diego, USA

Correspondence to: E. Marañón (em@uvigo.es)

**Abstract**

The deep chlorophyll maximum (DCM) is a ubiquitous feature of phytoplankton vertical distribution in stratified waters that is relevant for our understanding of the mechanisms that underpin the variability in photoautotroph ecophysiology across environmental gradients and has implications for remote sensing of aquatic productivity. During the PEACETIME (*Process studies at the air-sea interface after dust deposition in the Mediterranean Sea*) cruise, carried out from 10 May to 11 June 2017, we obtained 23 concurrent vertical profiles of phytoplankton chlorophyll *a*, carbon biomass and primary production, as well as heterotrophic prokaryotic production, in the western and central Mediterranean basins. Our main aims were to quantify the relative role of photoacclimation and enhanced growth as underlying mechanisms of the DCM and to assess the trophic coupling between phytoplankton and heterotrophic prokaryotic production. We found that the DCM coincided with a maximum in both biomass and primary production but not in growth rate of phytoplankton, which averaged 0.3 d$^{-1}$ and was relatively constant across the euphotic layer. Photoacclimation explained most of the increased chlorophyll *a* at the DCM, as the carbon to chlorophyll *a* ratio (C:Chl *a*) decreased from ca. 90-100 (g:g) at the surface to 20-30 at the base of the euphotic layer, while phytoplankton carbon biomass increased from ca. 6 mgC m$^{-3}$ at the surface to 10-15 mgC m$^{-3}$ at the DCM. As a result of photoacclimation, there was an uncoupling between chlorophyll *a*-specific and carbon-specific productivity across the euphotic layer. The fucoxanthin to total chlorophyll *a* ratio increased markedly with depth, as did the biomass contribution of large cells, suggesting a dominance of diatoms at the DCM. The increased biomass and carbon fixation at the base of the euphotic zone was associated with enhanced rates of heterotrophic prokaryotic activity, which also showed a surface peak linked with warmer temperatures. Considering the phytoplankton biomass and turnover rates measured at the DCM, nutrient diffusive fluxes across the nutricline were able to supply only a minor fraction of the photoautotroph nitrogen and phosphorus requirements. Thus the deep maxima in biomass and primary production were not fueled by new nutrients, but likely resulted from cell sinking from the upper layers in combination with the high photosynthetic efficiency of a diatom-rich, low-light acclimated community largely sustained by regenerated nutrients. Further studies with increased temporal and spatial resolution will be required to ascertain if the deep primary production peaks associated with the DCM persist across the western and central Mediterranean Sea throughout the stratification season.



# 1. Introduction

One of the most remarkable features of phytoplankton distribution in lakes and oceans is the presence of a deep chlorophyll maximum (DCM), typically located at the base of the euphotic layer and coinciding with the top of the nutricline, that occurs in permanently and seasonally stratified water columns (Cullen, 2015; Herbland and Voituriez, 1979). Multiple, non-mutually exclusive mechanisms may contribute to the development of a DCM, including photoacclimation (the increase in cellular chlorophyll content as a response to low light conditions) (Geider, 1987), enhanced growth conditions at the layer where elevated nutrient diffusion from below coexists with still sufficient irradiance (Beckmann and Hense, 2007), a decrease in sinking rates near the pycnocline (Lande and Wood, 1987), and changes in buoyancy regulation or swimming behaviour of cells (Durham and Stocker, 2012). Photoacclimation is a rapid process that takes place in a matter of hours (Fisher et al., 1996), and therefore part of the increased chlorophyll concentration at the DCM is always the result of a decrease in the phytoplankton carbon to chlorophyll $a$ ratio (C:Chl $a$), which results mainly from decreased irradiance but is also favoured by enhanced nutrient supply (Geider et al., 1996). Although the role of photoacclimation, particularly in strongly oligotrophic environments, has long been acknowledged (Steele, 1964), the fact that Chl $a$ is used routinely as a surrogate for photoautotrophic biomass has helped to fuel the assumption, often found in the scientific literature and in textbooks, that the DCM is always a maximum in the biomass and, by extension, the growth rate of phytoplankton. The assessment of total phytoplankton biomass along vertical gradients has been traditionally hindered by the time-consuming nature of microscopy techniques, but the increasing use of optical properties such as the particulate beam attenuation and backscattering coefficients to estimate the concentration of suspended particles in the water column (Behrenfeld et al., 2016; Martinez-Vicente et al., 2013) has allowed to characterize biogeographic and seasonal patterns in the vertical variability of phytoplankton chlorophyll and biomass in stratified environments (Fennel and Boss, 2003; Cullen, 2015; Mignot et al., 2014).

It is now established that the nature of DCM changes fundamentally along a gradient of thermal stability and nutrient availability (Cullen, 2015). In the oligotrophic extreme, represented by permanently stratified regions such as the subtropical gyres, the DCM is mostly a result of photoacclimation and does not constitute a biomass maximum (Marañón et al., 2000; Mignot et al., 2014; Pérez et al., 2006). However, a biomass maximum, located at a shallower depth than the DCM, can develop in oligotrophic conditions as a result of the interplay between phytoplankton growth, biological losses and sinking (Fennel and Boss, 2003). In mesotrophic regimes, such as seasonally stratified temperate seas during summer, the DCM is often also a biomass maximum that manifests as a peak in beam attenuation or backscattering (Mignot et al., 2014). Both ends of this trophic gradient can be found in the Mediterranean Sea along its well-known longitudinal trend in nutrient availability, phytoplankton biomass, and productivity (Antoine et al., 1995; D'Ortenzio and Ribera d'Alcalà, 2009; Lavigne et al., 2015). Using data from biogeochemical Argo (BGC-Argo) profiling floats deployed throughout the Mediterranean, Barbieux et al. (2019) established general patterns in the distribution and seasonal dynamics of biomass (estimated from the particulate backscattering coefficient) and chlorophyll subsurface maxima. They found that in the western Mediterranean Sea, during late spring and summer, a subsurface biomass maximum develops that coincides with a chlorophyll maximum and is located roughly at the same depth as the nutricline and above the 0.3 mol quanta $m^{-2}$ $d^{-1}$ isolume. In contrast, in the Ionian and Levantine seas the DCM, which has a smaller magnitude, arises solely from photoacclimation and is located well above the nutricline at a depth that corresponds closely with the 0.3 mol quanta $m^{-2}$ $d^{-1}$ isolume (Barbieux et al., 2019). The presence of a subsurface or deep biomass maximum may suggest that a particularly favourable combination of light and nutrients occurs at that depth, leading to enhanced phytoplankton growth and new production. It remains unknown, however,





whether phytoplankton growth and biomass turnover rates are actually higher at the depth of the biomass maximum. An additional source of uncertainty is that both the particulate attenuation and backscattering coefficients relate not only to

phytoplankton abundance but to the entire pool of particles, including non-algal and detrital particles, which are known to contribute significantly to total suspended matter in oligotrophic regions (Claustre et al., 1999). Combining direct and

specific measurements of phytoplankton production (with the [14]C-uptake technique) and biovolume (with flow cytometry) offers a way to determine photoautotrophic biomass turnover rates (Marañón et al., 2014; Kirchman, 2002)

and thus gain further insight into the dynamics and underlying mechanisms of the DCM. By investigating concurrently the vertical variability in heterotrophic prokaryotic production in relation to phytoplankton standing stocks and

productivity, it is also possible to ascertain potential implications of the DCM for trophic coupling within the microbial plankton community.

The PEACETIME (*Process studies at the air-sea interface after dust deposition in the Mediterranean Sea*) cruise, which investigated atmospheric deposition fluxes and their impact on biogeochemical cycling in the Mediterranean Sea

(Guieu et al., 2020), covered the Western, Tyrrhenian and Ionian regions during late spring 2017, when the DCM was already well developed. Here we describe the vertical variability in chlorophyll *a* concentration, phytoplankton biomass

and production, and heterotrophic prokaryotic production. Our main goals are: 1) to determine the extent to which photoacclimation, enhanced phytoplankton biomass, and enhanced productivity and growth underlie the DCM; 2) to

characterize the vertical variability in C:Chl *a*, and C biomass-specific and Chl *a*-specific production, and 3) to assess the trophic coupling between phytoplankton photosynthetic activity and heterotrophic bacterial production. The results

presented provide a context, in terms of the abundance and activity of key microbial plankton groups, to other ecological and biogeochemical investigations carried out during the PEACETIME cruise and included in this special

issue.

## 2. Methods

### 2.1 Oceanographic cruise

A detailed description of the ensemble of atmospheric and oceanographic observations conducted during the

PEACETIME process study can be found in Guieu et al. (2020). Here we report measurements conducted during an oceanographic cruise on board the *R/V Pourquoi Pas?*, which took place in the western and central Mediterranean Sea

during the period 10 May – 11 June 2017 (Fig. 1). The cruise focused on three long-stay stations, which were occupied during 4-5 days: station TYRR, located in the Tyrrhenian Sea (39° 20.4' N, 12° 35.6' E); station ION, located in the

Ionian Sea (35° 29.1' N, 19° 47.8' E); and station FAST, located in the Balearic Sea (37° 56.8' N, 2° 54.6' E). The latter station was occupied as part of a fast-action response to investigate the biogeochemical impacts of an event of

atmospheric wet deposition that occurred during the period 3-5 June (Guieu et al., 2020). In addition, 10 short-stay stations were occupied during 8 hours. At all stations, CTD casts were conducted and seawater samples obtained for the

measurement of the abundance, biomass and productivity of phytoplankton and bacterioplankton.

### 2.2 Sampling, hydrography and irradiance

We used a Seabird Electronics's SBE911+ CTD underwater unit interfaced with a sampling carousel of 24 Niskin bottles, a Chelsea Acquatracka 3 fluorometer and a photosynthetically active radiation (PAR) Biospherical Licor sensor.

At the short stations, CTD casts were conducted at 04:00-07:00 local time (with the exception of station 1, which was sampled at 08:40). At the long stations, CTD casts were conducted throughout the day but in the present report, to avoid

the effect of diel variability, we only consider plankton samples from the pre-dawn casts (04:00-05:00). Using CTD





casts conducted between 06:00 and 16:00, we calculated the value of the euphotic-layer vertical attenuation coefficient

($k_d$) after fitting the PAR data to:

$$PAR_z = PAR(0^-) \exp(-k_d z) \qquad (1)$$

where $PAR(0^-)$ is the irradiance just below the surface. From this model we calculated the % PAR level for each
sampling depth, which was used to determine the incubation irradiance for each sample during the primary production

experiments (see section 2.5 below). We compared the daily integrated values of total solar irradiance (TSI) from the
ship's pyranometer (Young 70721) and the theoretical incident PAR above the surface ($PAR(0^+)$) from the model of

Frouin et al. (1989) and used the highest ratio (corresponding to the clearest sky conditions encountered during the
cruise) to obtain a conversion factor (0.42) that transforms TSI into $PAR(0^+)$. TSI units (W m$^{-2}$) were converted to

photon flux units (mol quanta m$^{-2}$ s$^{-1}$) by multiplying by 4.6 and a PAR ($0^-$) to $PAR(0^+)$ ratio of 0.958 was applied
(Mobley and Boss, 2012). Using $k_d$ and $PAR(0^-)$ values for each sampling day the daily irradiance reaching each

sampling depth z was calculated with Eq. 1.

**2.3 Phytoplankton abundance and biomass**

The abundance of phytoplankton cells with an equivalent spherical diameter (ESD) below 5-6 μm was determined with
flow cytometry. Seawater samples (4.5 mL in volume) from 8-10 depths in the euphotic zone were fixed with

glutaraldehyde grade I (1% final concentration), flash-frozen with liquid nitrogen and stored at -80 °C until analysis.
Cell counts were performed on a FACSCanto II flow cytometer (Becton Dickinson). The separation of different

autotrophic populations (*Synechococcus*, picoeukaryotes and small nanophytoplankton) was based on their scattering
and fluorescence signals according to Marie et al. (2000) and Larsen et al. (2001). To obtain estimates of carbon

biomass, we applied different values of cellular carbon content for each group. For *Synechococcus*, we used a cell
carbon content of 0.15 pgC cell$^{-1}$, which is the mean value obtained by Buitenhuis et al. (2012) from a compilation of

multiple open-ocean studies. For picoeukaryotes, we assumed a mean cell diameter of 2 μm and thus a volume of 4.2
μm$^3$ cell$^{-1}$, which gives a carbon content of 0.72 pgC cell$^{-1}$ after applying the relationship between cell volume and cell

carbon obtained by Marañón et al. (2013) with 22 species of phytoplankton spanning 6 orders of magnitude in cell
volume. For small nanophytoplankton, we assumed a mean cell diameter of 4 μm and a volume of 34 μm$^3$ cell$^{-1}$, which

gives a carbon content of 4.5 pgC cell$^{-1}$.

The abundance of phytoplankton cells with an ESD above 6 μm was determined with an Imaging Flow CytoBot (IFCB)

(Olson and Sosik, 2007), which quantitatively images chlorophyll *a*-fluorescing particles. Samples (4.7 mL) were
obtained from 6-8 depths in the euphotic zone and screened through a 150-μm mesh to prevent clogging of the

instrument. From each obtained image phytoplankton biovolume was computed following Moberg and Sosik (2012).
Processed images, metadata, and derived morphometric properties were uploaded to EcoTaxa (https://ecotaxa.obs-

vlfr.fr/). The biovolume concentration was converted into a carbon biomass concentration by applying the mean carbon
to volume ratio obtained by Marañón et al. (2013) for cells larger than 6 μm in ESD (0.11 pgC μm$^{-3}$). Total

phytoplankton biomass was calculated as the sum of the carbon biomass of *Synechococcus*, picoeukaryotes,
nanoeukaryotes and > 6 μm phytoplankton.

**2.4 Pigments**

Samples for pigment analysis with high-performance liquid chromatography (HPLC) were collected from 12 depths

over the 0-250 m range. Depending on particle load, a volume of 2-2.5 L of seawater was vacuum-filtered under low





pressure onto Whatman GF/F filters (ca. 0.7 µm pore size, 25 mm in diameter). The filters were flash-frozen
immediately after filtration in liquid nitrogen, stored at -80° during the cruise and shipped back to the laboratory in
cryo-shipping containers filled with liquid nitrogen. Filters were extracted in 3 mL of pure methanol at -20°C for one
hour. The extracts were vacuum-filtered through GF/F filters and then analyzed (within 24 h) by HPLC using a
complete Agilent Technologies system. The pigments were separated and quantified following the protocol described in
Ras et al. (2008). Here we report the concentration of total chlorophyll *a* (TChl *a*), which includes chlorophyll *a* and
divinyl chlorophyll *a*. The fucoxanthin to TChl *a* ratio was multiplied by different factors to obtain estimates of the
diatom contribution to TChl *a*. The factors used were: 1.41 (Uitz et al., 2006), 1.6 (Di Cicco et al., 2017) and 1.74 (Di
Cicco, 2014).

**2.5 Primary production**

Primary production (PP) was measured with the $^{14}$C-uptake technique using simulated in situ incubations on deck. For
each sampling depth (5-6 depths distributed between 5 m and the base of the euphotic layer), seawater was transferred
from the Niskin bottle to 4 polystyrene bottles (3 light and one dark bottles) of 70 mL in volume, which were amended
with 20-40 µCi of NaH$^{14}$CO$_3$ and incubated for 24 h in on-deck incubators that were refrigerated with running seawater
from the ship's continuous water supply. The incubators were provided with different sets of blue and neutral density
filters that simulated the following percentages of attenuation: 70, 52, 38, 25, 14, 7, 4, 2 and 1%. We incubated the
samples at an irradiance level (% PAR) as close as possible to the one corresponding to their depth of origin. After
incubation, samples were filtered, using low-pressure vacuum, through 0.2-µm polycarbonate filters (47 mm in
diameter). At 3 depths on each profile (5 m, 15-30 m and the DCM), samples were filtered sequentially through 2-µm
and 0.2-µm polycarbonate filters, thus allowing to determine primary production in the picophytoplankton (< 2 µm) and
the nano- plus micro-phytoplankton (> 2 µm) size classes. All filters were exposed to concentrated HCl fumes
overnight, to remove non-fixed, inorganic $^{14}$C, and then transferred to 4-mL plastic scintillation vials to which 4 mL of
scintillation cocktail (Ultima Gold XR) were added.

We also measured dissolved primary production at 3 depths on each profile (surface, base of the euphotic layer and an
intermediate depth), following the method described in Marañón et al. (2004) but using the same incubation bottles
employed to determine particulate primary production. Briefly, after incubation one 5-mL aliquot was taken from each
incubation bottle and filtered through a 0.2-µm polycarbonate filter (25 mm in diameter), using low-pressure vacuum.
Filters were processed as described above, whereas the filtrates were acidified with 100 µL of 5M HCl and maintained
in an orbital shaker for 12 hours. Then, 15 mL of liquid scintillation cocktail were added to each sample. The
radioactivity in all filter and filtrate samples was measured on-board with a Packard 1600TR liquid scintillation counter.
The percentage of extracellular release (% PER) was calculated as dissolved primary production divided by the sum of
dissolved and particulate primary production.

To calculate daily PP, DPM counts in the dark samples were subtracted from the DPM counts in the light samples and
actual values of dissolved inorganic carbon concentration, determined during the cruise at each sampling depth, were
used. Given that all incubations were conducted at SST, we applied a temperature correction to the measured rates, by
using the Arrhenius-van 't Hoff equation:

$$R = A\,e^{\,Ea/KT} \qquad\qquad (2)$$

where R is the production rate, A is a coefficient, Ea is the activation energy, K is the Boltzmann's constant (8.617 10$^{-5}$
eV °K$^{-1}$) and T is temperature in °K. The value of production rate obtained for each sampling depth incubated at SST





was used to determine A, and then R was calculated for the in situ temperature at each sampling depth. Following Wang et al. (2019), we used a value of Ea = 0.61 eV, which corresponds approximately to a $Q_{10}$ value of 2.3. The turnover rate of phytoplankton biomass (growth rate, $d^{-1}$) was calculated by dividing the rate of production (mgC $m^{-3}$ $d^{-1}$) by the concentration of phytoplankton carbon (mgC $m^{-3}$) (Kirchman, 2002).

**2.6 Heterotrophic prokaryotic production**

Heterotrophic prokaryotic production (BP) was estimated from rates of $^{3}$H-Leucine incorporation using the microcentrifugation technique (Smith and Azam, 1992) as detailed in Van Wambeke et al. (2020). Briefly, triplicate 1.5-mL samples and one blank from 10 depths between surface and 250 m were incubated in the dark in two thermostated incubators set at 18.6°C for upper layers and 15.2°C for deeper layers. Leucine was added at 20 nM final concentration and the Leucine to carbon conversion factor used was 1.5 kg C $mol^{-1}$. Given that in situ temperature varied from 13.4 to 21.6°C, temperature corrections were applied by using a $Q_{10}$ factor determined on two occasions during the cruise, when different samples were incubated simultaneously in the two incubators. We obtained two values of $Q_{10}$ (3.9 and 3.3), from which an average value of 3.6 was used for the whole BP data set. The same $Q_{10}$ was applied to assess the contribution of temperature to the variability of BP in the upper water column, by comparing BP at in situ temperature and at a constant temperature of 17°C.

**3. Results**

**3.1 Hydrographic conditions**

All three long stations were characterized by broadly similar values of sea surface temperature (SST) (20-21°C) and strong thermal stratification, with a 5-6°C thermocline extending over the 10-70 m depth range (Fig. 2a). Compared to TYRR, stations ION and FAST showed warmer SST and a stronger stratification, and station FAST presented the warmest subsurface waters. The short stations covered a wider range of locations and consequently exhibited higher variability in SST and in the strength and vertical extent of the thermocline (Fig. S1). Throughout the cruise, nutrient concentrations were low (< 0.5 μmol $L^{-1}$ for nitrate and < 0.03 μmol $L^{-1}$ for phosphate) in the upper 50-60 m of the water column (Guieu et al., 2020). The nitracline, defined as the first depth where nitrate concentration exceeded 0.5 μmol $L^{-1}$, was located at (mean ± SD) 71 ± 3, 105 ± 2 and 78 ± 8 m in stations TYRR, ION and FAST, respectively. The phosphacline, defined as the first depth where phosphate concentration exceeded 0.03 μmol $L^{-1}$, was deeper: 86 ± 3, 181 ± 7 and 90 ± 5 at TYRR, ION and FAST, respectively (Table 1). At all stations, fluorescence profiles displayed a DCM (see section 3.2) which was located approximately at the 1% PAR depth and 5-10 m above the 0.3 mol $m^{-2}$ $d^{-1}$ isolume (Fig. 2b, Fig. S1, Table 1). Both the DCM depth and the 1% PAR depth were shallower at station TYRR (74 ± 4 and 71 ± 8, respectively) than at station ION (96 ± 4 and 94 ± 6, respectively), with station FAST showing intermediate values (Fig. 2b, Table 1). The depths of both the nitracline and the phosphacline were strongly correlated with the DCM depth throughout the cruise (Pearson's $r$ = 0.86, $n$ = 23, $p$ < 0.001 for the nitracline depth and $r$ = 0.74, $n$ = 23, $p$ < 0.001 for the phosphacline depth).

**3.2 Phytoplankton total chlorophyll $a$, biomass and production**

Surface total chlorophyll $a$ concentration (TChl $a$) was low (≤ 0.1 mg $m^{-3}$) throughout most of the cruise (Fig. 3a,b,c; Fig. S2a), with the only exception of short station 1, which sampled a filament of enhanced phytoplankton abundance (Fig. 1). The mean surface TChl $a$ was similar in all three long stations (0.07-0.08 mg $m^{-3}$). All vertical profiles displayed a marked DCM (Fig. 3a,b,c; Fig. S2a), with peak TChl $a$ values in the range 0.4-0.7 mg $m^{-3}$ at stations TYRR





and ION and 0.4-1.0 mg m$^{-3}$ at station FAST. The mean DCM TChl *a* at the three stations was similar (0.6 mg m$^{-3}$) (Table 1). Vertically integrated (from surface to the euphotic layer depth) TChl *a* was higher and more variable at FAST

(21 ± 9 mg m$^{-2}$) compared with TYRR (16 ± 2 mg m$^{-2}$) and ION (18 ± 2 mg m$^{-2}$) (Table 1).

Phytoplankton carbon biomass tended to increase with depth, exhibiting maxima at either intermediate depths (40-50

236 m) or at the base of the euphotic layer (80-100 m) (Fig. 3d,e,f; Fig. S2b). The concentration of phytoplankton C in surface waters was relatively invariant at 6 mgC m$^{-3}$ whereas mean biomass values at the DCM in stations TYRR, ION

and FAST were 13 ± 8, 11 ± 1 and 16 ± 10 mgC m$^{-3}$, respectively (Table 1). Thus the increase, from the surface to the base of the euphotic layer, in phytoplankton biomass was ca. 2-fold, compared with ca. 8-fold for TChl *a*. Comparing

the deep to surface ratios in TChl *a* and C biomass in the three stations indicates that increased phytoplankton biomass was responsible for 22-34 % of the increased TChl *a* at the DCM, while photoacclimation (decreased C:Chl *a* at depth)

was responsible for the remaining 66-78 %.

Compared to surface values, the deep maxima in phytoplankton C biomass were of smaller magnitude than those of

244 TChl *a*. Consequently, the mean C:Chl *a* ratio (g:g) was much higher at the surface (89-97) than at the DCM (21-34) at all long stations (Table 1). Considering together the data from all stations, C:Chl *a* increased with light availability

following a saturating curve (Fig. 4a). Particulate primary production (PP) ranged between 1 and 3 mgC m$^{-3}$ d$^{-1}$ in surface waters, and tended to increase with depth (Fig. 3h,i,j; Fig. S2c). In most profiles (19 out of 23), the highest

value of PP (typically, 3-6 mgC m$^{-3}$ d$^{-1}$) was measured in the deepest sample, corresponding to the DCM. There were only small differences in mean integrated PP among stations, which ranged between 170 ± 36 and 209 ± 67 mgC m$^{-2}$ d$^{-1}$

at TYRR and FAST, respectively (Table 1).

The contribution of cells larger than 2 μm in diameter (nano and micro-phytoplankton) to total phytoplankton biomass

increased with depth from ca. 60 % at the surface to ca. 80 % at the base of the euphotic layer, taking an overall, mean value of 68 ± 13 % for all samples pooled together (Fig. S3a). The contribution of the > 2 μm size class to total PP was

relatively stable both among stations and with depth, taking a mean value of 73 ± 6 % in the long stations (Fig. S3b). In contrast, the percentage of extracellular release (PER) showed a marked vertical pattern in all stations, decreasing with

depth from a mean value of 42 ± 8 % at the surface to 22 ± 4 % at the DCM (Fig. S3c).

TChl *a*-specific primary production (P$^{Chl}$) displayed a marked light dependence, following a saturating function of light

availability and reaching values of 20-35 mgC mgChl *a*$^{-1}$ d$^{-1}$ at near-surface irradiance levels (Fig. 4b). In contrast, the ratio between primary production and phytoplankton C biomass (P$^{C}$, equivalent to a biomass turnover rate) was

independent of irradiance (Pearson's *r* = 0.17, *n* = 77, *p* = 0.14), with most values falling within the range 0.1-0.5 d$^{-1}$ throughout the euphotic layer (Fig. 4c). Overall, the mean P$^{C}$ for the whole cruise was 0.3 ± 0.1 d$^{-1}$ and the same mean

P$^{C}$ (0.3 d$^{-1}$) was measured in the surface and the DCM.

### 3.3 Fucoxanthin to total chlorophyll *a* ratio

The fucoxanthin to total chlorophyll *a* ratio (Fuco:TChl *a*) consistently increased below the upper 40-50 m in all long stations (Fig. 5). Fuco:TChl *a* mean values at the surface were 0.036 ± 0.001 at TYRR, 0.040 ± 0.004 at ION and 0.051

± 0.005 at FAST (Table 1). Using different conversion factors (see Methods), these ratios translate into a range of diatom contribution to TChl *a* of 5-6 %, 6-7 % and 7-9 % at TYRR, ION and FAST, respectively. At the DCM,

Fuco:TChl *a* was 0.21 ± 0.04 at TYRR, 0.29 ± 0.03 at ION and 0.24 ± 0.10 at FAST, which corresponds to diatom contributions of 30-36 %, 41-51 % and 34-42 %, respectively.





**3.4 Heterotrophic prokaryotic production and relationship with primary production**

Rates of heterotrophic prokaryotic production (BP) in the euphotic layer fell within the range 10-50 ngC L$^{-1}$ h$^{-1}$ and took values < 10 ngC L$^{-1}$ h$^{-1}$ in the waters below (Fig. 6, Fig. S4). Most vertical profiles of BP were characterized by two peaks: one at the surface and another one in sub-surface waters, coinciding with the DCM or slightly above it. We assessed the effect of temperature on BP rates in the upper layer (0-50 m) by comparing the rates calculated at in situ temperature versus a constant temperature of 17°C (mean temperature for all profiles across 0-250 m). While BP rates at in situ temperature displayed a marked increase in the 2-3 most shallow sampling depths, BP at a constant temperature of 17°C remained largely homogenous with depth (Fig. S5). The mean integrated BP at the long stations ranged between 50-60 mgC m$^{-2}$ d$^{-1}$ in stations TYRR and ION and ca. 90 mgC m$^{-2}$ d$^{-1}$ in station FAST (Table 1). Considering all data in the euphotic layer, there was a positive correlation between both particulate and dissolved primary production and BP (Fig. 7). However, primary production explained less than 10% of the variability in BP. Taking into account that BP displayed a surface maximum, which was rarely observed in the primary production profiles, we explored the relationship between PP and BP in samples from below 30 m (Fig. S6). Although a positive relationship was observed, PP still explained only a small amount of variability in BP, which reflects the fact that the deep maximum in BP was often shallower than the deep PP maximum.

**4. Discussion**

**4.1 Seasonal and geographical context**

The vertical location and longitudinal variability of the DCM we observed agree with the patterns previously reported for the Mediterranean Sea, based both on climatological analyses of chlorophyll *a* profiles (Lavigne et al., 2015) and time-series studies (Marty et al., 2002; Lemée et al., 2002). In the western basin, where the spring bloom is characterized by the presence of a surface chlorophyll maximum, a subsurface maximum develops from April onwards that takes progressively a deeper location, reaching 70-80 m in mid-summer. This deepening of the DCM occurs later in the north than in the south section of the western basin (Lavigne et al., 2015). In agreement, we found during PEACETIME that the stations located in the southwest had deeper DCMs than those located in the northwest (it has to be noted, though, that the seasonal evolution during the cruise may have influenced the DCM depth and that the southwestern stations were sampled last). In the central Mediterranean (e.g. Ionian Sea), the spring surface chlorophyll maximum does not occur, and the DCM also appears around April but becomes deeper than in the western region. Accordingly, during our cruise the DCM at long station ION was significantly deeper than at the western stations. We also found, as previously described in analyses of vertical structure in stratified waters (Herbland and Voituriez, 1979; Cullen, 2015; Letelier et al., 2004), a general correspondence between the top of the nutricline and the depth of the DCM, with deeper values in the Ionian Sea than in the western basin. These differences reflect the more persistent stratification and stronger degree of oligotrophy that characterizes the central and eastern basins as compared to the western Mediterranean Sea (Bosc et al., 2004; D'Ortenzio and Ribera d'Alcalà, 2009).

Numerous surveys at fixed stations (Lemée et al., 2002; Marty and Chiavérini, 2002) as well as along oceanographic transects (Estrada, 1996; Moutin and Raimbault, 2002; López-Sandoval et al., 2011) have described the vertical variability of PP in the Mediterranean Sea during the stratification season. While subsurface maxima are often observed in late spring and summer, these peaks tend to be located above rather than at the DCM (Estrada, 1996; Marty and Chiavérini, 2002). During the MINOS cruise, which sampled the entire Mediterranean Sea from the western to the Levantine basin in May-June 1996, Moutin and Raimbault (2002) found a strong correlation between the depths of the





deep PP peak and the DCM depth, but the former was on average 20 m shallower. In contrast, during PEACETIME the

mean depths of the primary production maximum and the DCM coincided and only on 3 profiles was the primary

production peak located the DCM. One potential source of bias during our $^{14}$C-uptake experiments could come

from the fact that all samples were incubated at sea surface temperature. However, the correction we applied to the

measured rates assumes a relatively strong degree of temperature dependence (an activation energy of 0.66 eV), while

oligotrophic conditions, prevailing during the cruise, are known to result in decreased temperature sensitivity of

phytoplankton metabolic rates (O'Connor et al., 2009; Marañón et al., 2018). Had we used a lower temperature

sensitivity in our corrections, the magnitude of the deep production peaks would have been even greater. Thanks to the

combined measurements of cell abundance and biovolume together with photosynthetic carbon fixation, it is possible to

explore the variability in phytoplankton biomass and its turnover rate, to assess if the measured deep production peaks

are plausible and explore which processes may have been responsible for their occurrence (section 4.2).

Our estimates of growth rate also allowed us to assess if phytoplankton inhabiting the surface waters of the

Mediterranean Sea during the stratification season were just experiencing nutrient limitation of their standing stock

(yield limitation sensu Liebig) or if they are also limited in their rate of resource use (physiological rate limitation sensu

Blackman). As demonstrated in chemostat experiments (Goldman et al., 1979), fast growth rates are compatible with

extremely low ambient nutrient concentrations and therefore oligotrophy in itself does not necessarily imply that

Blackman limitation is operating. However, the mean growth rate measured in surface waters during the PEACETIME

cruise (0.3 d$^{-1}$) is well below the maximal, nutrient-saturated growth rate that could be expected at warm (> 20°C)

temperatures for different groups such as diatoms, cyanobacteria and green algae ($\geq$ 1 d$^{-1}$) (Kremer et al., 2017).

Similarly low (0.2-0.6 d$^{-1}$) phytoplankton growth rates have been reported before for the western Mediterranean Sea

(Pedrós-Alió et al., 1999), the Atlantic subtropical gyres (Marañón, 2005) and recently the North Pacific subtropical

gyre (Berthelot et al., 2019). Multiple experimental approaches, including in situ iron additions (Boyd et al., 2007;

Yoon et al., 2018) in high-nutrient, low-chlorophyll regions as well as in vitro bioassays with nutrients (Mills et al.,

2004; Tanaka et al., 2011; Tsiola et al., 2016) and desert dust (Marañón et al., 2010; Guieu et al., 2014) in low-nutrient,

low-chlorophyll regions, typically display larger increases in carbon fixation and nutrient uptake rates than in

photoautotroph abundance, which implies enhanced biomass turnover rates upon alleviation of nutrient scarcity.

Therefore low nutrient availability, which is widespread in the global ocean (Moore et al., 2013), results not only in low

phytoplankton biomass but also in slow growth rates.

**4.2 Mechanisms underlying deep production maxima**

Earlier studies have shown that both photoacclimation and enhanced biomass contribute to the occurrence of the DCM

in the western Mediterranean Sea whereas in the central and eastern basins photoacclimation alone would be mainly

responsible for the increased chlorophyll *a* at depth (Barbieux et al., 2019; Estrada, 1996; Mignot et al., 2014). In

contrast, during our survey the contribution of increased phytoplankton biomass was similar in all stations, including

the one located in the Ionian Sea. Most (ca. 75%) of the increased Chl *a* concentration at the DCM at all stations was

due to photoacclimation, the rest being a result of increased biomass. The C:Chl *a* ratios (g:g) we estimate

(approximately 90-100 and 20-30 for surface and DCM populations, respectively) agree well with previous results from

the Mediterranean Sea (Estrada, 1996) and the Atlantic subtropical gyres (Veldhuis and Kraay, 2004; Marañón, 2005;

Pérez et al., 2006) as well as with general patterns observed in light- and nutrient-limited laboratory cultures (MacIntyre

et al., 2002; Halsey and Jones, 2015; Behrenfeld et al., 2016). The fact that high C:Chla values (> 50) persisted

throughout the water column until PAR was lower than 2 mol m$^{-2}$ d$^{-1}$ suggests that nutrient limitation prevailed over



most of the euphotic layer, because under nutrient-sufficient and light-limited conditions C:Chl *a* typically takes values
< 30 (Halsey and Jones, 2015). Only the populations inhabiting the DCM showed clear signs of light limitation,
reflected in the decreased C:Chl *a* ratios. The question remains whether those populations were mainly sustained by
new nutrients supplied by diffusion from below the nutricline or by recycled nutrients originated within the euphotic
layer.

Taillandier et al. (2020) combined measurements of the vertical gradient in nutrient concentrations during PEACETIME
with estimates of diffusivity based on turbulent kinetic energy dissipation rates measured by Ferron et al. (2017) in the
western Mediterranean Sea, which allowed them to calculate the vertical diffusive fluxes across the nutricline in the
Tyrrhenian Sea and the Algerian Basin. We used these fluxes to estimate the contribution of new nutrients to sustain
phytoplankton productivity at the deep biomass maximum in stations TYRR and FAST, given the observed biomass
concentration and turnover rate (Table 2) and assuming that the deep biomass maximum extended over 30 m. These
calculations suggest that diffusive fluxes could provide only a small fraction of the nitrogen and, especially, the
phosphorus requirements of the phytoplankton assemblages inhabiting the lower part of the euphotic layer. Thus most
of the primary production in the euphotic layer was sustained by recycled nutrients, which agrees with the observation
that phytoplankton growth rates did not show any increase at the DCM despite the proximity of the nutricline. The
broadly homogeneous distribution of phytoplankton growth throughout the euphotic layer also supports the conclusion
of Fennel and Boss (2003) that deep phytoplankton maxima develop approximately at the compensation depth, where
growth and losses balance each other. We can speculate that the compensation depth during our cruise broadly
coincided with the 1% PAR light level or 0.5 mol m$^{-2}$ d$^{-1}$ isolume but additional primary production measurements in
deeper samples would have been required to test this hypothesis.

The nano- and micro-phytoplankton size classes consistently dominated primary production during the cruise,
accounting on average for ca. 70% of total carbon fixation. The relatively low share (≤ 30-35 %) of primary production
due to picophytoplankton agrees well with previous results based on remote sensing across the entire Mediterranean Sea
(Uitz et al., 2012) while field measurements conducted in the western and central basins during the stratification season
show somewhat higher and more variable picophytoplankton contribution (Magazzù and Decembrini, 1995;
Decembrini et al., 2009). During PEACETIME, the contribution of cells > 2 μm in diameter and cells > 5 μm in
diameter to total phytoplankton biomass increased with depth, and this trend was associated with a significant increase
in the contribution of diatoms to total phytoplankton biomass, which reached at least 30 % in the DCM of all stations,
and was particularly high (nearly 50 %) in the most stratified station, located in the Ionian Sea. Deep maxima in diatom
abundance are common in the Mediterranean Sea during stratified conditions (Ignatiades et al., 2009; Siokou-Frangou
et al., 2010; Mena et al., 2019) and are often associated with peaks in biogenic silica (Crombet et al., 2011). The
increased prevalence of diatoms at the base of the euphotic layer, which illustrates the ecological diversity of this group
(Kemp and Villareal, 2018), is likely a result of multiple adaptations and mechanisms, including high growth efficiency
under low light conditions (Fisher and Halsey, 2016), buoyancy regulation (Villareal et al., 1996), the ability to exploit
transient nutrient pulses through luxury uptake and storage (Kemp and Villareal, 2013; Cermeño et al., 2011) and the
enhanced ammonium assimilation mediated by microbial interactions in the phycosphere (Olofsson et al., 2019).

### 4.3 Phytoplankton photophysiology and productivity

Although the widespread occurrence of deep chlorophyll maxima, which cannot be detected by ocean colour sensors, is
often mentioned as a shortcoming of satellite-based productivity models, the vertical distribution of chlorophyll *a*
concentration can be derived from surface optical properties (Uitz et al., 2006; Morel and Berthon, 1989). The key



challenge rests in the quantification of the photophysiological parameters (e.g. photosynthetic efficiency), required to convert photoautotroph biomass or pigment concentration into a measure of carbon fixation. Of especial relevance, in the case of low-light acclimated populations, is the initial slope in the relationship between irradiance and Chl $a$-specific photosynthesis ($\alpha^B$, mgC (mgChl $a$)$^{-1}$ h$^{-1}$ (µmol photon m$^{-2}$ s$^{-1}$)$^{-1}$). Using a large dataset of photosynthetic parameters obtained with the same method, Uitz et al. (2008) found that $\alpha^B$ took a mean value of $0.025 \pm 0.022$ in the lower part of the euphotic layer in oligotrophic regions across the world's oceans. Assuming 14 h of daylight and that night-time respiration losses account for 20 % of carbon fixed during the day (Geider, 1992), and given the mean Chl $a$ concentration (0.6 mg m$^{-3}$) and daily PAR (0.5 mol m$^{-2}$ d$^{-1}$) measured at the DCM during PEACETIME, this value of $\alpha^B$ translates into a primary production < 1.7 mgC m$^{-3}$ d$^{-1}$, lower than the rates we measured (2-10 mgC m$^{-3}$ d$^{-1}$). Interestingly, the mean $\alpha^B$ value determined at the base of the euphotic layer during the PROSOPE cruise, which sampled all major basins of the Mediterranean Sea in September 1999, was $0.066 \pm 0.024$, which would correspond to a DCM primary production of 4.4 mg C m$^{-3}$ d$^{-1}$, in agreement with our observations. The low $\alpha^B$ value reported by Uitz et al. (2008) largely reflected the photophysiological properties of communities dominated by small cells, in contrast with the assemblages encountered during the present study. It thus would appear that the high primary production at the DCM during PEACETIME was due not only to enhanced levels of phytoplankton biomass but also to the presence of a diatom-rich community characterised by high photosynthetic efficiency. These results stress the importance of incorporating the linkage between community structure and photophysiological parameters to improve the application of bio-optical productivity models over diverse ecological and biogeographic settings (Robinson et al., 2018; Uitz et al., 2010; Uitz et al., 2012).

We found that phytoplankton can sustain similar rates of biomass-specific carbon fixation across a wide range of irradiances, in spite of considerable variations in Chl $a$-specific photosynthesis. The uncoupling between these two metrics of productivity likely arises from photoacclimation, whereby cells receiving less irradiance invest more resources in light-harvesting complexes and are thus capable of sustaining similar rates of nutrient-limited carbon fixation (per unit biomass) as cells experiencing high light availability (Pan et al., 1996). Using a photoacclimation model in conjunction with satellite observations of phytoplankton carbon and Chl $a$, Behrenfeld et al. (2016) demonstrated that most of the seasonal and interannual variability in surface Chl $a$ concentration of multiple ocean biomes resulted from photoacclimation and therefore cannot be readily translated into equivalent changes in productivity. Our results suggest that the same conclusion also applies to small-scale vertical variability in stratified environments, where phytoplankton growth rates are relatively constant across the euphotic layer (Pérez et al., 2006; Berthelot et al., 2019). More generally, the fact that C:Chl $a$ is highly sensitive not only to irradiance but to nutrient availability and temperature as well (Geider, 1987; Halsey and Jones, 2015) means that changes in growth rate can be disconnected from Chl $a$-specific photosynthesis across multiple environmental gradients (Cullen et al., 1992; Marañón et al., 2018).

**4.4 Relationship between heterotrophic prokaryotic and primary production**

The vertical distribution of BP, which was characterized by the presence of both surface and deep maxima, likely reflects the combined influence of several controlling factors. Different studies have investigated the relationship between temperature, inorganic nutrients and dissolved organic matter availability as drivers of heterotrophic prokaryotic production and carbon demand in the Mediterranean Sea over seasonal (Céa et al., 2015; Lemée et al., 2002; Alonso-Sáez et al., 2008) and mesoscale to basin-scale (Pulido-Villena et al., 2012; Pedrós-Alió et al., 1999) ranges of variability but the relative role of these factors at the small vertical scale within the upper water column has





been comparatively less explored. Van Wambeke et al. (2002) reported that BP consistently peaked at the surface

during a mesoscale survey in the Gulf of Lions in spring, which was probably a result of the fact that primary

production also increased in the surface layer, a pattern also reported by Lemée et al. (2002) throughout most of the

432    year in the DYFAMED station. In the case of the PEACETIME cruise, however, the surface peak in BP cannot be

attributed to increased primary production, which took the lowest values in the surface layer. Temperature, which

exhibited a ca. 5°C-gradient over the upper 50 m, appears as the most likely responsible driver of the surface BP peaks,

considering that the estimated rates at a constant temperature of 17°C were nearly homogeneous across the upper layer.

Seasonal studies in coastal waters of the western Mediterranean Sea have also identified temperature as a factor that

contributes to explain the temporal variability of bacterial production in surface waters (Alonso-Sáez et al., 2008; Céa et

al., 2015). In contrast, the deep peak in BP found during our cruise was associated, at least in part, with increased

phytoplankton biomass and production, so an enhanced availability of organic substrates may have been responsible for

the stimulation of bacterial activity near the base of the euphotic layer.

Atmospheric deposition of nutrients may have also contributed to sustain the surface BP peaks observed during our

study. Nitrogen and phosphorus amendments to seawater from the mixed layer resulted in BP stimulation after 48 h,

indicating NP co-limitation of BP, whereas addition of a labile carbon source (glucose) had no effect (Van Wambeke et

al., 2020). Thus the surface BP peak observed under in situ conditions was not due to dependence of organic carbon

substrates but may have resulted in part from new N and P availability through dry atmospheric deposition. The same

study shows that atmospheric dry deposition during the PEACETIME cruise could sustain about 13 % of the

heterotrophic bacterial N demand within the mixed layer. Other sources of N fueling heterotrophic bacteria could come

from recycling, as for instance hydrolysis of proteins satisfied a mean of 47% of that demand (Van Wambeke et al.,

2020).

Despite the association between increased PP and increased BP in subsurface waters, the overall strength of the

relationship between these two variables during PEACETIME was weak. This in contrast with previous analyses in the

Mediterranean Sea that included a much broader range of plankton biomass and production regimes than the one

covered during our cruise and found stronger correlations between photosynthetic carbon fixation and BP (Turley et al.,

2000; Pulido-Villena et al., 2012). If we consider the trophic coupling between heterotrophic bacteria and

phytoplankton as the extent to which dissolved primary production meets heterotrophic bacterial carbon demand

(Morán et al., 2002), our results suggest a poor coupling during the PEACETIME cruise. Assuming a value of bacterial

growth efficiency of 10 %, as determined in the western Mediterranean Sea during summer (Alonso-Sáez et al., 2008;

Lemée et al., 2002), our measured rates of dissolved primary production represented, on average, only 25 % (SD = 14

%) of estimated bacterial carbon demand. Similar weak phytoplankton-bacterioplankton coupling has been reported

before for the Mediterranean Sea during the stratification period (Alonso-Sáez et al., 2008; López-Sandoval et al., 2011;

Morán et al., 2002), which emphasizes the role of additional substrates, other than recent products of photosynthesis

released in dissolved form, in fuelling bacterial metabolism. These additional substrates can include dissolved organic

carbon released by consumers (e.g. sloppy feeding) or during cell lysis, as well as organic molecules previously

produced and accumulated over time scales longer than 1 day or derived from allochthonous sources such as river and

atmospheric inputs. However, the fact that bacterial carbon demand often exceeds the instantaneous rate of dissolved

primary production does not mean that bacterial metabolism is independent of phytoplankton photosynthesis over

annual scales, but rather reflects the temporal uncoupling resulting from the episodic nature of phytoplankton

production events (Karl et al., 2003; Steinberg et al., 2001; Morán and Alonso-Sáez, 2011).





### 4.5 Conclusions

We have shown that the DCM in the western Mediterranean Sea during the stratification period, already known to be a phytoplankton biomass maximum, can also represent a substantial primary production maximum. These deep maxima in biomass and primary production are not associated with an increase in phytoplankton growth rates and do not seem to be fueled by new nutrients, but likely arise as a result of cell sinking from above in combination with the high photosynthetic efficiency of a diatom-rich, low-light acclimated community, which sustains similar growth rates as those measured in the upper, well-illuminated layers. Because of the variability in C:Chl *a* ratios, changes in Chl *a*-specific primary production can be disconnected from biomass turnover rates. While the trophic coupling between heterotrophic bacteria and phytoplankton was relatively poor, the increased photosynthetic biomass and carbon fixation measured near the base of the euphotic zone did result in an enhancement of bacterial heterotrophic activity, which in the surface layers appeared to be regulated by temperature. Our results support the combined use of isotope uptake measurements and biovolume-based estimates of phytoplankton carbon biomass to derive growth rates at discrete depths and gain insight into the mechanisms underlying the DCM. Data with higher spatial and temporal resolution, as derived for instance from optical sensors attached to autonomous instruments, will allow to establish if the marked peaks in primary production we observed are a persistent feature of the DCM in the central and western Mediterranean Sea, and to quantify their broader biogeochemical significance.

### Data availability

All data from the PEACETIME cruise (https://doi.org/10.17600/17000300) are stored at the LEFE CYBER Database (http://www.obs-vlfr.fr/proof/php/PEACETIME/peacetime.php) and will be made freely available once all manuscripts are submitted to the PEACETIME special issue. In the meantime, data can be also obtained upon request to the first author.

### Acknowledgements

This study is a contribution to the PEACETIME project (http://peacetime-project.org), a joint initiative of the MERMEX and ChArMEx components supported by CNRS-INSU, IFREMER, CEA, and Météo-France as part of the programme MISTRALS coordinated by INSU. The research of EM and MPL was funded by the Spanish Ministry of Science, Innovation and Universities through grant PGC2018-094553B-I00 (POLARIS) awarded to EM. We acknowledge the French Centre National d'Etudes Spatiales (CNES), which supported the bio-optical and ocean color component of the PEACETIME project (PEACETIME-OC). We also thank the *Service d'analyse de pigments par HPLC* (SAPIGH) at the Institut de la Mer de Villefranche (IMEV) for HPLC analysis, as well as the captain and crew of the R/V *Pourquoi Pas?* for their help during the work at sea.

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



**Table 1.** Mean and standard deviation (in brackets) for different physical, chemical and biological variables at the three long stations. Nitracline depth is the first depth at which nitrate concentration reached 0.5 µmol L$^{-1}$ while phosphacline

depth corresponds to the first depth at which phosphate concentration reached 0.03 µmol L$^{-1}$. Chlorophyll *a* concentration and particulate primary production (PP) were integrated from the surface to the 1% PAR depth.

Heterotrophic prokaryotic production (BP) was integrated from the surface to 200 m.

| Variable | TYRR | ION | FAST |
|---|---|---|---|
| Surface temperature (°C) | 20.1 (0.6) | 20.4 (0.1) | 21.4 (0.2) |
| Surface TChl *a* (mg m$^{-3}$) | 0.07 (0.01) | 0.07 (0.01) | 0.08 (0.01) |
| Nitracline depth (m) | 71 (3) | 105 (2) | 78 (8) |
| Phosphacline depth (m) | 86 (3) | 181 (7) | 90 (5) |
| DCM depth (m) | 74 (4) | 96 (4) | 85 (6) |
| 1% PAR depth (m) | 71 (8) | 94 (6) | 81 (5) |
| 0.3 mol m$^{-2}$ d$^{-1}$ isolume depth (m) | 80 (7) | 104 (5) | 91 (6) |
| PAR at DCM (mol m$^{-2}$ d$^{-1}$) | 0.47 (0.26) | 0.45 (0.06) | 0.44 (0.19) |
| DCM TChl *a* concentration (mg m$^{-3}$) | 0.57 (0.11) | 0.57 (0.07) | 0.62 (0.29) |
| Surface phytoplankton biomass (mgC m$^{-3}$) | 6 (1) | 6 (1) | 6 (2) |
| DCM phytoplankton biomass (mgC m$^{-3}$) | 13 (8) | 11 (1) | 16 (10) |
| Surface C:Chl *a* ratio (g:g) | 97 (8) | 91 (5) | 89 (23) |
| DCM C:Chl *a* ratio (g:g) | 27 (10) | 21 (1) | 34 (8) |
| Surface Fucoxanthin:TChl *a* ratio | 0.036 (0.001) | 0.040 (0.004) | 0.051 (0.005) |
| DCM Fucoxanthin:TChl *a* ratio | 0.21 (0.04) | 0.29 (0.03) | 0.24 (0.10) |
| Integrated TChl *a* (mg m$^{-2}$) (0 - 1% PAR z) | 16 (2) | 18 (2) | 21 (9) |
| Integrated PP (mgC m$^{-2}$ d$^{-1}$) (0 - 1% PAR z) | 170 (36) | 186 (56) | 209 (67) |
| % integrated PP > 2 µm (0 - 1% PAR z) | 72 (4) | 75 (6) | 73 (3) |
| Integrated BP (mgC m$^{-2}$ d$^{-1}$) (0 - 200 m) | 57 (3) | 51 (9) | 89 (10) |





**Table 2.** Estimation of the contribution of nutrient diffusive fluxes to sustain the requirements of the deep phytoplankton biomass maximum (DBM) in stations TYRR and FAST. The DBM layer considered has a thickness of 30 m and the nutrient requirements of primary production are assumed to follow Redfield C:N:P proportions. The magnitude of nitrate and phosphate diffusive fluxes at the base of the DBM is taken from Taillandier et al. (2020).

|  | TYRR | FAST |
| --- | --- | --- |
| Mean phytoplankton concentration (mgC m$^{-3}$) | 15 | 10 |
| Biomass turnover rate (d$^{-1}$) | 0.3 | 0.3 |
| C:N molar ratio of phytoplankton biomass | 6.6 | 6.6 |
| C:P molar ratio of phytoplankton biomass | 106 | 106 |
| Vertical extent of DBM layer (m) | 30 | 30 |
| Lower limit of deep biomass layer (m) | 60 | 80 |
| N requirement of DBM ($\mu$mol N m$^{-2}$ d$^{-1}$) | 1705 | 1136 |
| P requirement of DBM ($\mu$mol P m$^{-2}$ d$^{-1}$) | 107 | 71 |
| Diffusive N flux (Taillandier et al. 2020) ($\mu$mol N m$^{-2}$ d$^{-1}$) | 560 | 101 |
| Diffusive P flux (Taillandier et al. 2020) ($\mu$mol P m$^{-2}$ d$^{-1}$) | 12.8 | 2.3 |
| % of N requirement met by diffusive flux | 33 | 9 |
| % of P requirement met by diffusive flux | 12 | 3 |


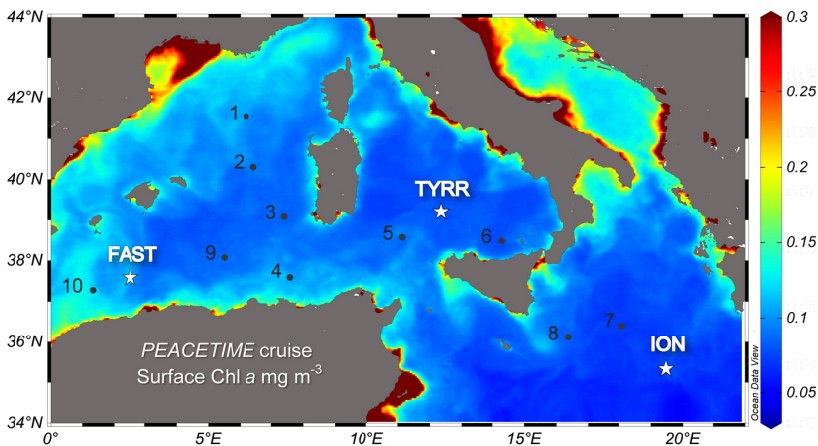

**Figure 1.** Location of the sampled stations superimposed on a map of ocean colour-based surface chlorophyll *a*
concentration (mg m$^{-3}$) averaged over the period of the PEACETIME cruise (12 May – 8 June 2017). Dots and stars
indicate the location of short and long stations, respectively. Ocean colour data from MODIS/Aqua, NASA Goddard
Space Flight Center.

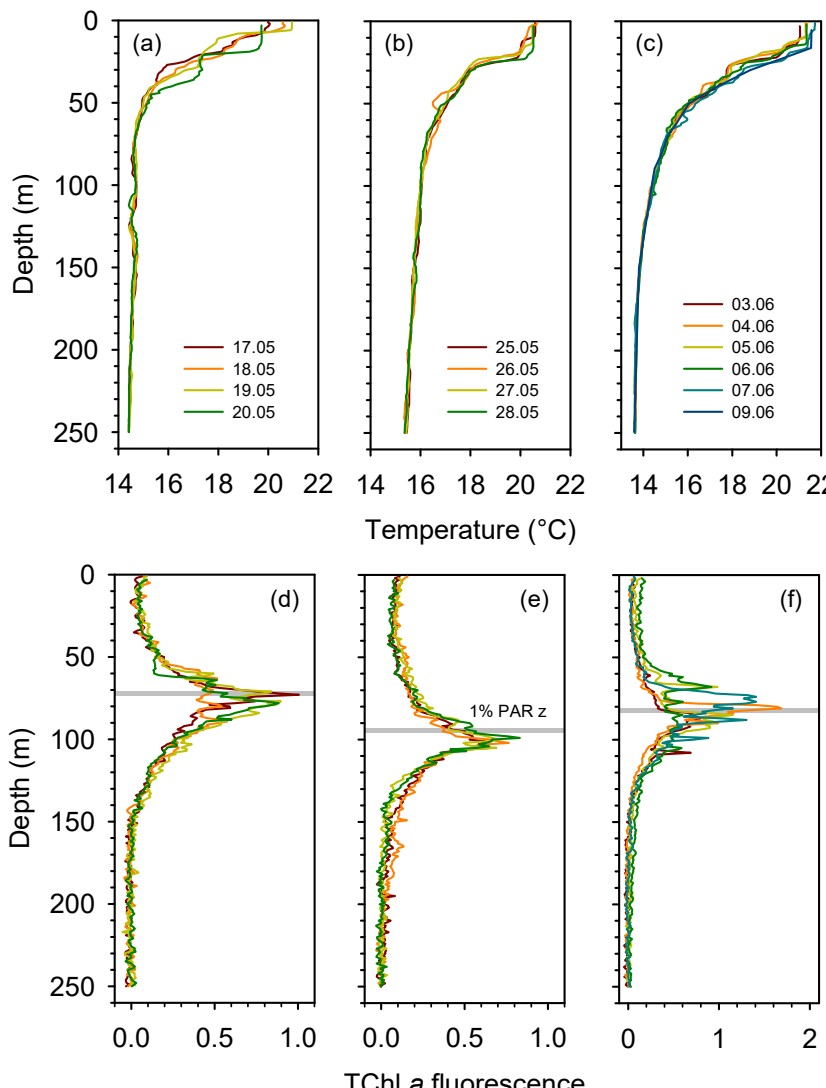

**Figure 2.** Vertical profiles of temperature and fluorescence (0-250 m) during each sampling day at the long stations TYRR (a, c), ION (b, e) and FAST (c, f). The colour code denotes the sampling date in dd.mm format, and the grey bars indicate the mean value of the 1% PAR depth at each station. The fluorescence signal was calibrated against HPLC-determined total chlorophyll $a$ concentration.





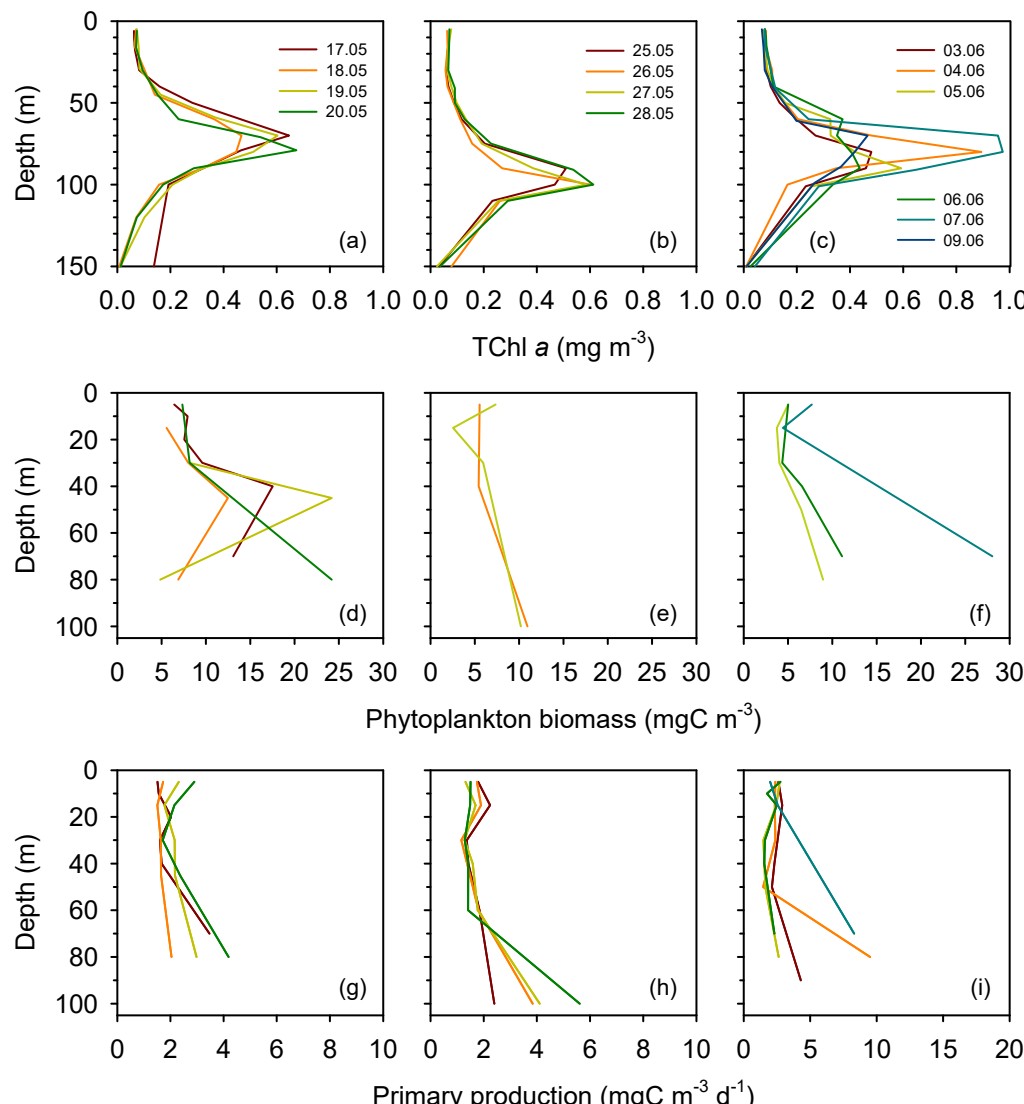

**Figure 3.** Vertical profiles of total chlorophyll *a* concentration (a,b,c), phytoplankton biomass concentration (d,e,f) and primary production (g,h,i) during each sampling day at the long stations TYRR (a,d,g), ION (b,e,h) and FAST (c,f,i).

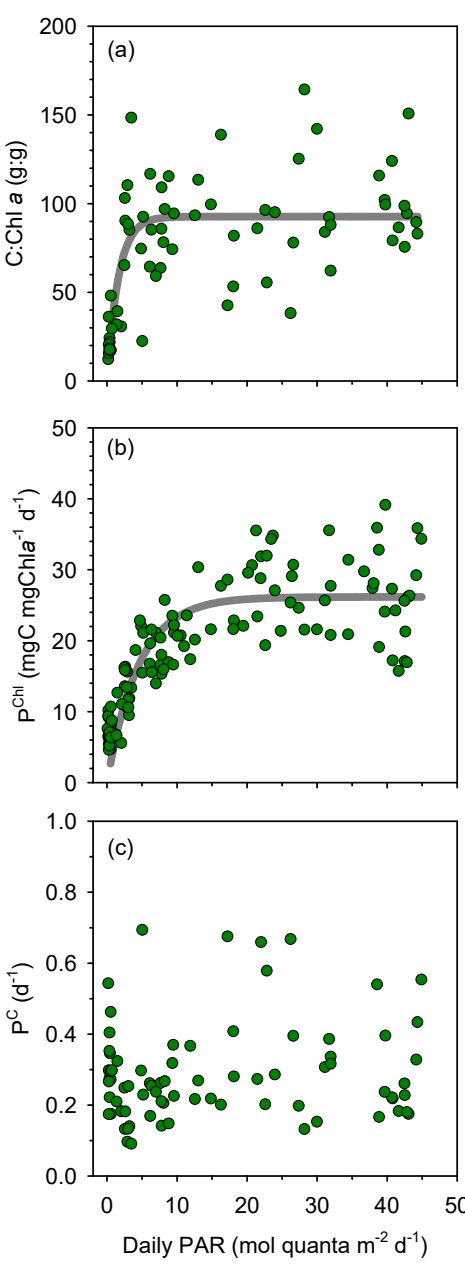

**Figure 4**. Relationship between PAR and a) phytoplankton carbon to chlorophyll *a* ratio, b) chlorophyll *a*-specific
particulate primary production and c) phytoplankton biomass turnover rate with data from all stations pooled together.
The non linear fits are (a) $y = 92.7 (1 - \exp(-0.61 x))$, $r^2 = 0.53$, $p < 0.001$, $n = 69$ and (b) $y = 26.2 (1 - \exp(-0.22 x))$,
$r^2 = 0.68$, $p < 0.001$, $n = 119$.



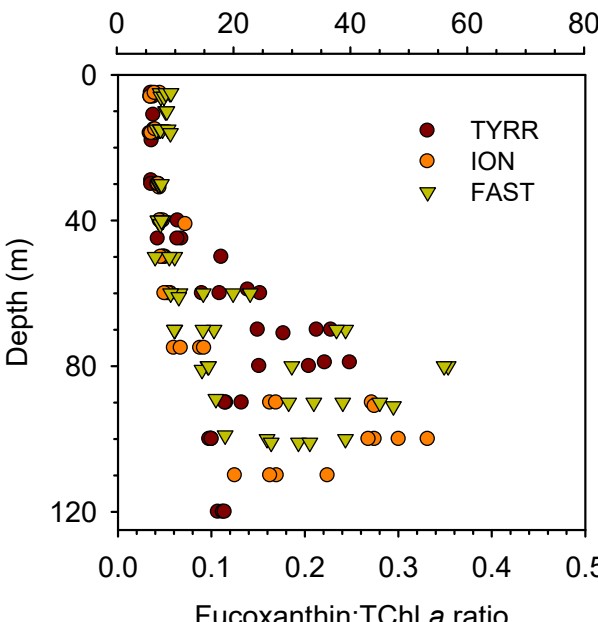

**Figure 5.** Vertical variability of the fucoxanthin to total chlorophyll *a* concentration ratio in the three long stations. The
upper *x*-axis is included as a reference and shows the estimated diatom contribution to TChl *a* computed with the mean
value of three different conversion factors. See Methods for details.





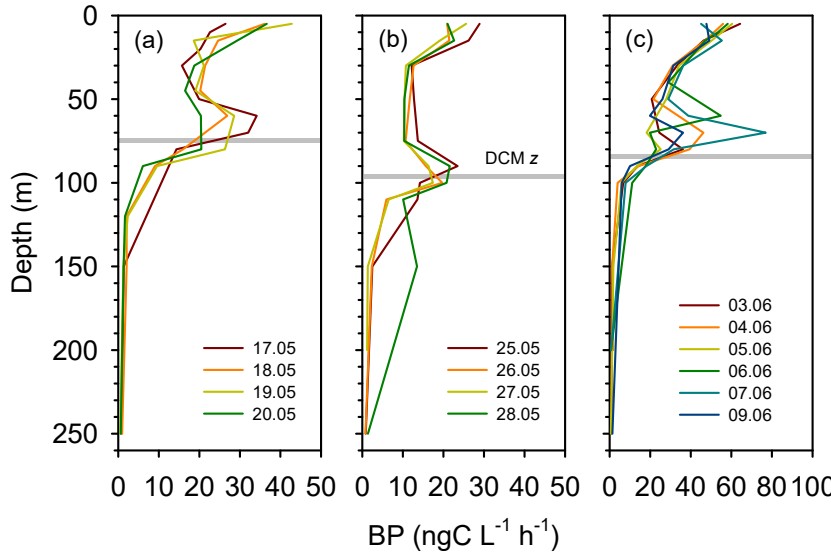

**Figure 6.** Vertical profiles of heterotrophic prokaryotic production (BP, dawn casts only) during each sampling day at
the long-term stations a) TYRR, b) ION and c) and FAST. The grey line indicates depth of the DCM at each station.

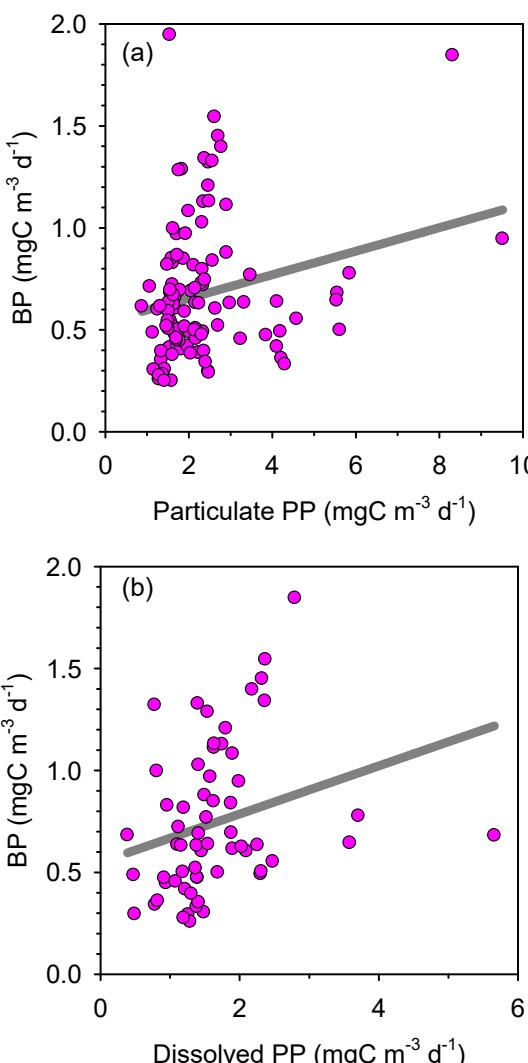

824

**Figure 7.** Bacterial production as a function of a) particulate and b) dissolved primary production with data from all stations pooled together. The linear regression models are (a) $y = 0.058 \, x + 0.54$ ($r^2 = 0.05$, $n = 110$, $p = 0.016$) and (b) $y = 0.12 \, x + 0.55$ ($r^2 = 0.07$, $n = 62$, $p = 0.034$).

828