# Peer review of "Deep maxima of phytoplankton biomass, primary production and bacterial production in the Mediterranean Sea"

_Biogeosciences, 2020_

## Referee Comment (RC1) · Anonymous Referee #1 · 24 Aug 2020

This is an interesting paper in which the authors discuss the role of photoacclimation and enhanced growth as the underlying mechanism of the DCM in the Mediterranean Sea during the late spring. The study was carried out from 10 May to 11 June during the PEACETIME cruise.

The study is exciting; however, I have the following comments/suggestions which will make this manuscript publishable after authors incorporate and modify the paper.

How can you be sure that the dominance of diatoms at the DCM was resulted from cell sinking from the upper layers due to photoacclimation rather than the new production?

I suggest the authors to check any physical mechanism like the role of Rossby wave

etc. Analysis of the physical processes in the region is compulsory when you discuss the DCM properties and the underlying mechanism.

At stations TYRR and FAST, DCM was deeper than nitracline depth. However, DCM was located above the nitracline depth at ION. From this, I understand that the physical processes operating at ION may be different from the other two stations. Hence, insisting to look into the water column stability in all the three stations during the measurement period.

In all the three stations, DCM was just below the 1% PAR depth and below the nitracline depth except the station ION; where nitracline depth was deeper than DCM. Have you noticed any difference in phytoplankton characteristics in the DCM at ION compared to the other two? I feel you can make out the difference from the size of the phytoplankton cell. Please check it and confirm that your hypothesis is true in all the three stations.

It is also not clear how the individual role of photoacclimation and biomass contribution was explored? Please mention the way to quantify it?

I feel that the manuscript needs major revision by addressing the above comments.

―――――――――――――――――――――――

---

## Referee Comment (RC2) · Anonymous Referee #2 · 11 Sep 2020

This work reports that the deep chlorophyll maximum (DCM) is a maximum of biomass and primary production in the oligotrophic Mediterranean Sea during late spring. These deep maxima are accompanied by a sub-maximum of bacterial production. The ms is relevant, it reveals that primary production is very significant at the DCM, a component of production undetectable by remote sensing techniques. It is worth mentioning that the biomass data presented are quite new, since the biomass of picoplankton and especially of nanoplankton, the latter seldom directly quantified, were analyzed with specific and appropriate techniques. The ms is well organized and well written and is very easy to read. The figures and tables are clear and explanatory. The results may represent a challenge for some current paradigms of phytoplankton ecophysiology. The main

factors that regulate phytoplankton growth rates are light, nutrients and temperature. The study concludes that growth rates remain more or less constant along the water column. Between the surface layers and the DCM, irradiance decreases from saturating to limiting conditions and temperature decreased about 5 C in this study. These two factors alone should have significantly decreased phytoplankton growth rates at the DCM, which could have been compensated somehow by an increase of diffusive nutrient supply to the DCM from the nutricline. However, the measured nutrient supply was low. The authors explain their findings by the presence of a diatom community in the DCM layer that was very efficient at low irradiances (I would add temperature). The implications would be important since these results show that composition conditions the phytoplankton response, which should question general ecophysiological assumptions that are often extrapolated to natural conditions by some models. The following are some issues that I suggest be examined further to reinforce the important findings of the study (sentences copied from the ms are signaled between quotation marks)

Carbon estimates Estimates of C biomass are paramount in this work. More accurate biovolume estimates can be obtained using the scattering properties (forward or side scattering) of single cells than by assuming mean volumes for picoplankton and nanoplankton. In addition, this procedure would take into account the important changes of cell size with depth, often ignored (Binder et al. 1996. Dynamics of pico-phytoplankton, ultraphytoplankton and bacteria in the central equatorial Pacific. Deep. Res. II 43: 907-931, Mena et al 2019, cited by the authors). Please, specify the volume analyzed for detecting a significant number of cells from the small nanophytoplankton fraction, it is an interesting information that can help other researches and future studies. L. 138. "Thus the increase, from the surface to the base of the euphotic layer, in phytoplankton biomass was ca. 2-fold, compared with ca. 8-fold for TChl a." Please, consider recalculating the biomass taking into account changes of biovolume with depth.

Diatoms at the DCM L. 264. "The fucoxanthin to total chlorophyll a ratio (Fuco: TChl a)

consistently increased below the upper 40-50 m in all long stations." From the changes in this ratio it is deduced that diatom contribution increases with depth. Fucoxanthin is also present in haptophytes and pelagophytes, two main components of phytoplankton with 19'hex-fuco and 19'but-fuco as their main diagnostic pigments, respectively. To make sure the increase in fucoxanthin is due to diatoms I would recommend calculating the vertical distribution of fucoxanthin: (19'hex-fuco + 19'but-fuco). The increase of this ratio with depth would be a more convincing evidence of a differential increase in diatoms. The images obtained with the Imaging Flow CytoBot should help to confirm that diatoms dominated or were very abundant in the DCM layer.

L. 375. "...this trend was associated with a significant increase in the contribution of diatoms to total phytoplankton biomass, which reached at least 30 % in the DCM of all stations, and was particularly high (nearly 50 %) in the most stratified station, located in the Ionian Sea." Please, re-check your estimates of diatom contribution at the DCM. Although I agree that diatoms can increase at the DCM, these values appear very high. In addition, the data of Crombet et al 2011 (cited in ms) show a patchy distribution of the Deep Silica Maximum and diatoms in the DCM of the Mediterranean.

Primary production (PP) at the DCM L. 309. "In contrast, during PEACETIME the mean depths of the primary production maximum and the DCM coincided and only on 3 profiles was the primary production peak located above the DCM." The subsequent discussion does not present potential mechanisms to explain the discrepancy in PP estimates at the DCM between this and previous studies cited in the ms, which show a PP maximum above the DCM most of the time. It does argue that the high primary production at the DCM during PEACETIME was due not only to enhanced levels of phytoplankton biomass but also to the presence of a diatom-rich community character-ized by high photosynthetic efficiency. It is a bit surprising that the same response has not been found in previous studies in the area. Could it be possible that the presence of diatoms with high photosynthetic efficiency at the DCM discussed by the authors is a consequence of the previous spring bloom at the surface and not a regular feature

of the DCM in the Mediterranean? Estrada et al (1993. Variability of deep chlorophyll maximum characteristics in the Northwestern Mediterranean. Mar. Ecol. Prog. Ser. 92: 289–300) reported the occasional presence of diatoms from a decaying bloom that contributed significantly to the DCM biomass but with a very low photosynthetic efficiency, which seems typical of sinking cells. It seems that a large contribution of diatoms in the DCM layer is not a general feature of the Mediterranean Sea, and perhaps could explain the discrepancies in PP estimates at this depth with other studies.

L. 340. "In contrast, during our survey the contribution of increased phytoplankton biomass was similar in all stations, including the one located in the Ionian Sea." An important conclusion is that DCM is a maximum of biomass and production in the Mediterranean, at least during the period of the study. However, in 3 of the 4 profiles obtained in the Tyrrhenian Sea the biomass maximum is well above the DCM. This result is mentioned (line 235) but ignored throughout the ms. Moreover, it is difficult to explain how the PP maximum can be found at 70-80m, at the DCM and below the biomass maxima in these stations without a significant increase of nutrient supply. The correction that has been applied to short-term temperature variations to estimate PP at in-situ temperature from incubations at higher temperatures (about 5 C) could be discussed further to see if they may have distorted the results of the deep layers.

L. 444. "Thus the surface BP (bacterial production) peak observed under in situ conditions was not due to dependence of organic carbon substrates but may have resulted in part from new N and P availability through dry atmospheric deposition." This explanation can be applied to phytoplankton as well. If atmospheric input of inorganic nutrients and recycling are the main reasons for vertical patterns of bacterial production, the same pattern should have been found for primary production (which is the pattern usually found by other studies in the Mediterranean cited in the ms).

L. 335. "Therefore low nutrient availability, which is widespread in the global ocean (Moore et al., 2013), results not only in low phytoplankton biomass but also in slow growth rates." This conclusion is controversial in the scientific community. Another

line of research with direct estimates of growth rates using mainly dilution experiments argue that, even with low nutrient concentrations, fast supply of nutrients from recycling results in the predominance of phytoplankton, usually of small size, with relatively high growth rates (Laws, E. A., 2013. Evaluation of in situ phytoplankton growth rates: A synthesis of data from varied approaches. Ann. Rev. Mar. Sci. 5: 247–268, and ref therein), although lower than those of taxa typical for bloom situations. Different optimal growth rates can be a function of taxonomical affiliation or size, among other reasons.

Keep the same y-scale for fig 3g, h and i.

END OF REVIEW

---

## Referee Comment (RC3) · Anonymous Referee #3 · 28 Oct 2020

This MS addresses the ubiquitous subsurface feature, the deep chlorophyll maximum, phytoplankton biomass and production, and heterotrophic prokaryotic production in the Mediterranean sea's stratified water column during the later spring season (May 10, 2017- June 11, 2017). This subsurface feature in the world ocean is known for long, more prominent in waters of lower latitude, are often found at nutracline depth well below the remote sensing reach, thus supports the importance of seaboard measurements to capture this feature. Chlorophyll a, an indicator of the phytoplankton biomass, is regulated by light, nutrient, etc. Here, the authors mainly aim to quantify photoacclimation's relative role and enhanced growth as an essential DCM mechanism. Secondly, the trophic coupling between phytoplankton and heterotrophic prokaryotic production is also addressed. Based on shipboard measurements in the Mediterranean sea, authors conclude that the DCM located at subsurface depth coincides with both biomass and primary production but not in growth rate and explains that the photoacclimation process leads to the increased chlorophyll a at the DCM. This study contributes vital insight into likely future ocean changes under the ocean warming scenario, thus merits publication of this work. However, I do not recommend a journal publishing this work in the present form. A few concerns about the methodology and the data interpretation need to be taken care of before considering this work for publication (see below).

Flow cytometry tool followed to obtain estimates of the carbon biomass in different size categories does not seem to have taken account of the autotrophic cells >150 microns in size neither their contribution is quantified, if minimal. The authors could have easily viewed these samples (>150micron) under the microscope to support the finding. If this were a significant observed, increased carbon biomass from the surface to the euphotic layer base would have been different and could lead to a different conclusion. The authors need to take care of this part in the section result and subsequently draw a conclusion at the end of the discussion section. Also, it is unclear whether definite size beads were run on flow cytometry to conclude the mean cell diameter used for carbon calculation. It is essential to show the reader the error introduced by assuming the mean cell diameter (2um or 4 um or 6um). On the other hand, while calculating Fucoxanthin to total chlorophyll a ratio calculation, I find authors have ignored that besides diatoms, Phaeocystis spp. are also potential sources of Fucoxanthin (see Latasa and Bidigare, 1998) instead accounted to diatom community. Furthermore, the presence of divinyl chl a, a marker for prochlorophyte, seems to have ignored and accounted for diatoms. Suggest authors revisit HPLC based pigment (depth-wise) analyses to rule out Prochlorococcus community is not missed out. In my opinion, low light-adapted Prochlorococcus at the DCM may be sizably contributing to the DCM community.

In my opinion, this manuscript (MS) needs revision in context to the points discussed above in the second paragraph before considering this paper for publication.

---

## Author Comment (AC1) · 9 Nov 2020

**Comments by Anonymous Referee #1 and our responses**

This is an interesting paper in which the authors discuss the role of photoacclimation and enhanced growth as the underlying mechanism of the DCM in the Mediterranean Sea during the late spring. The study was carried out from 10 May to 11 June during the PEACETIME cruise. The study is exciting; however, I have the following comments/suggestions which will make this manuscript publishable after authors incorporate and modify the paper.

We are grateful to the reviewer for their time and helpful comments.

How can you be sure that the dominance of diatoms at the DCM was resulted from cell sinking from the upper layers due to photoacclimation rather than the new production? I suggest the authors to check any physical mechanism like the role of Rossby wave etc. Analysis of the physical processes in the region is compulsory when you discuss the DCM properties and the underlying mechanism.

We have estimated the diffusive nutrient flux from below the DCM and found that it contributed a small fraction of the nutrient supply required to sustain the deep phytoplankton biomass maximum (see Table 2). In addition, we found no evidence of enhanced phytoplankton growth at the DCM. Hence our conclusion that the DCM was likely to result mainly from cell sinking from above, in a mechanism that has been modelled by Fennel and Boss (2003). The role of photoacclimation is supported not only by the strong decrease in C:Chla ratios at the DCM but also by the fact that our measurements of PP at the DCM imply higher photosynthetic efficiencies than commonly measured at the DCM in oligotrophic regions where small cells dominate (Uitz et al. 2008).

Planetary Rossby waves have been shown to uplift the DCM (Kawamiya and Oschlies 2001) and cause enhanced surface chlorophyll a values (Cippollini et al. 2001) but they are large-scale features that propagate westward over entire ocean basins. Topographic Rossby waves have been observed in the Mediterranean Sea although they are bottom-intensified fluctuations, and therefore not good candidates to explain the occurrence of the DCM. Following the reviewer's advice (see also comment below), we have explored the potential role of other physical mechanisms in originating the DCM. For instance, cells may accumulate in the vertical region of enhanced stability associated with the pycnocline. To explore this possibility, we calculated the depth of maximum Brunt-Väisälä frequency at each long station. We found that the layer of maximum stability lies at a depth of 15-25 m, well above the DCM, which does not support a role for this mechanism during the Peacetime cruise. In the revised version of the manuscript, we will add these data to Table 1.

At stations TYRR and FAST, DCM was deeper than nitracline depth. However, DCM was located above the nitracline depth at ION. From this, I understand that the physical processes operating at ION may be different from the other two stations. Hence, insisting to look into the water column stability in all the three stations during the measurement period.

Throughout the cruise, the depth of the DCM and the depth of the nitracline covaried. For the long stations, these two depths differed, on average, by less than 10 m. Indeed the nitracline was deeper than the DCM (on average, by 9 m) at ION whereas the opposite was true in the other long stations, but the actual difference was subtle and it is not clear that it means a different mechanism for DCM formation, particularly in view of the fact that we do not have information on the seasonal dynamics but rely on snapshot observations conducted during a 1-month cruise. Key properties such as the magnitude of the DCM, and the C:Chla ratio, phytoplankton growth rate and mean PAR at the DCM were the same at all three stations (see Table 1). As mentioned above, we have also looked into the depth of the maximum Brunt-Väisälä frequency, which was the same at ION and FAST (23 m). The fact that the nitracline was deeper than the DCM at ION probably reflects longitudinal differences in the way the DCM and the

mixed layer depth are coupled in the Mediterranean Sea, as discussed by Barbieux et al. (2019). These authors concluded, from the analysis of seasonal variability in the DCM using Biogeochemical Argo floats data, that in the Ionian and Levantine basins the deepest winter mixed layer rarely reaches the top of the nutricline and the DCM is persistently well above the nutricline during the stratified season. In the revised version of the manuscript, we will refer to this feature in the first section of the Discussion.

In all the three stations, DCM was just below the 1% PAR depth and below the nitracline depth except the station ION; where nitracline depth was deeper than DCM. Have you noticed any difference in phytoplankton characteristics in the DCM at ION compared to the other two? I feel you can make out the difference from the size of the phytoplankton cell. Please check it and confirm that your hypothesis is true in all the three stations. It is also not clear how the individual role of photoacclimation and biomass contribution was explored? Please mention the way to quantify it?

We have calculated, for the three long stations, the mean values of different variables that help to characterize phytoplankton size structure at the DCM: mean cell biovolume (from the imaging flow cytobot), % contribution of cells >6 µm to total phytoplankton C, and % contribution of cells > 2 µm to total primary production (from size-fractionated PP experiments). This information will be added to Table 1 in the revised manuscript. For all variables, we found no differences between stations, which suggests that, in terms of size structure, the DCM phytoplankton community was comparable among sites. The HPLC pigment data also indicated that in all three stations the contribution of diatoms at the DCM increased markedly in comparison with surface waters, as reflected in the fucoxanthin to total chlorophyll a ratio as well as the fucoxanthin to 19'hex-fuco+19'but-fuco ratio. These data will also be added to the revised version of the manuscript as a new figure in the supplementary information. The HPLC pigment data did suggest some differences among stations in the upper layers. For instance, within the upper mixed layer of ION and TYRR the phytoplankton assemblage was dominated by prymnesiophytes, followed by cyanobacteria, wheres the opposite was the case at FAST.

To estimate the contribution of photoacclimation (increased Chl per unit C biomass) and increased biomass to explain the DCM, we calculated the DCM to surface ratios for chlorophyll a and phytoplankton C concentration, as explained in section 3.2 of the manuscript. For instance, at station TYRR the deep to surface chla concentration ratio was 8.14 (0.57/0.07), while the deep to surface ratio in phytoplankton C was 2.17. This means that 26% (2.17/8.14) of the increased chl a at the DCM resulted from enhanced biomass, while the rest of the increased chl a (74%) resulted from photoacclimation. Repeating the same calculations for the other two stations indicates that, overall, photoacclimation accounted for 66-78% of the increase in chl a concentration from the surface to the DCM.

**References (not cited in the ms)**

Cippollini et al. (2001) Rossby waves detected in global ocean colour data. Geophys Res Lett 28:323-326

Kawamiya and Oschlies (2001) Formation of a basin-scale surface chlorophyll pattern by Rossby waves Geophys Res Lett 28:4139-4142

---

## Author Comment (AC2) · 9 Nov 2020

**Comments by Anonymous Referee #2 and our responses**

This work reports that the deep chlorophyll maximum (DCM) is a maximum of biomass and primary production in the oligotrophic Mediterranean Sea during late spring. These deep maxima are accompanied by a sub-maximum of bacterial production. The ms is relevant, it reveals that primary production is very significant at the DCM, a component of production undetectable by remote sensing techniques. It is worth mentioning that the biomass data presented are quite new, since the biomass of picoplankton and especially of nanoplankton, the latter seldom directly quantified, were analyzed with specific and appropriate techniques. The ms is well organized and well written and is very easy to read. The figures and tables are clear and explanatory.

We are grateful to this reviewer for their time and helpful comments.

The results may represent a challenge for some current paradigms of phytoplankton ecophysiology. The main factors that regulate phytoplankton growth rates are light, nutrients and temperature. The study concludes that growth rates remain more or less constant along the water column. Between the surface layers and the DCM, irradiance decreases from saturating to limiting conditions and temperature decreased about 5 C in this study. These two factors alone should have significantly decreased phytoplankton growth rates at the DCM, which could have been compensated somehow by an increase of diffusive nutrient supply to the DCM from the nutricline. However, the measured nutrient supply was low. The authors explain their findings by the presence of a diatom community in the DCM layer that was very efficient at low irradiances (I would add temperature). The implications would be important since these results show that composition conditions the phytoplankton response, which should question general ecophysiological assumptions that are often extrapolated to natural conditions by some models. The following are some issues that I suggest be examined further to reinforce the important findings of the study (sentences copied from the ms are signaled between quotation marks)

The observation that phytoplankton growth rates were rather invariant across the euphotic layer does seem counterintuitive in the face of strong gradients in irradiance and temperature. However, the same pattern (i.e. similar growth rates in the surface layer and near the base of the euphotic layer) has often been reported by other studies, such as (cited in the ms) Pérez et al. (2006) and Berthelot et al. (2019) and also (not cited in the ms) Cáceres et al. (2013), Landry et al. (2004) and Armengol et al. (2019). Specifically, Cáceres et al. (2013) found virtually the same growth rate throughout the euphotic layer in a station located in the eastern subtropical North Atlantic (their Fig. 7, bottom panels). Landry et al. (2004) measured the same growth rate at the surface and at 60 m in an offshore oligotrophic station off Southern California (their Fig. 2, Cruise P0605 Cycle 5). Armengol et al. 2019 reported (their Table 2, stations 1-7) a mean growth rate of $0.28 \pm 0.18$ $d^{-1}$ at the surface compared with $0.21 \pm 0.07$ $d^{-1}$ at the DCM in the central tropical Atlantic. In the revised version of the manuscript, we will add a reference to these additional studies in the Discusion (section 4.3).

The paradox of relatively constant phytoplankton growth throughout the euphotic layer in oligotrophic settings can perhaps be explained by considering that the physiological effect of a given environmental factor tends to decrease when another factor is limiting. Most laboratory experiments are designed to determine the effect of a single environmental driver while keeping other variables under optimal levels. For instance, under nutrient-sufficient conditions the effect of irradiance on growth is strong, and under optimal nutrient and irradiance conditions the effect of temperature is also strong. However, the temperature dependence of phytoplankton growth is greatly reduced under conditions of light (Edwards et al. 2016) or nutrient (Marañón et al. 2018) limitation. Conversely, the effect of increasing nutrient supply on growth is modest when temperatures are strongly limiting (see review by Cross et al. 2015). Thus the lack of irradiance effects on the growth rate of acclimated phytoplankton assemblages may result from the

fact that nutrient limitation prevails throughout the water column. We will add this suggestion to section 4.3 in the revised version of the manuscript.

Carbon estimates Estimates of C biomass are paramount in this work. More accurate biovolume estimates can be obtained using the scattering properties (forward or side scattering) of single cells than by assuming mean volumes for picoplankton and nanoplankton. In addition, this procedure would take into account the important changes of cell size with depth, often ignored (Binder et al. 1996. Dynamics of picophytoplankton, ultraphytoplankton and bacteria in the central equatorial Pacific. Deep. Res. II 43: 907-931, Mena et al 2019, cited by the authors).

Although estimates of cell biovolume based on the side scattering (SSC) signal were not routinely available for the cruise, we have examined a few profiles of SSC per cell at the long stations to assess depth-related changes in cell biovolume of *Synechococcus*, picoeukaryotes and nanoeukaryotes. As shown in the plots and table below, we found that cell biovolume of nano- and pico-eukaryotes decreased with depth whereas the opposite was true for *Synechococcus*.

[Figure]

Mean (and standard deviation) of the side-scatering signal per cell of different groups in the surface layer (0-40 m) and at the DCM (including also the sample obtained immediately above the DCM) in the three long stations (data from all stations were pooled together).

| | SSC per cell | | |
|---|---|---|---|
| | Nanoeukaryotes | Picoeukaryotes | *Synechococcus* |
| Surface (n = 13) | 1.06 (0.11) | 0.24 (0.04) | 0.052 (0.010) |
| DCM (n = 11) | 0.64 (0.15) | 0.21 (0.03) | 0.074 (0.017) |

We have recalculated the total biomass of phytoplanton at the DCM taking into account these depth-related changes in cell volume of pico and small nanophytoplankton. Specifically, the value of C biomass per cell used in the original calculations was multiplied by the observed DCM to surface SSC ratio for each group, which was 0.61 for nanoeukaryotes, 0.87 for picoeukaryotes and 1.42 for *Synechococcus*. The figure below shows that taking into account these changes in cell volume with depth has negligible effects on the estimated total phytoplankton biomass at the DCM:

[Figure]

Please, specify the volume analyzed for detecting a significant number of cells from the small nanophytoplankton fraction, it is an interesting information that can help other researches and future studies.

Samples were run at a flow rate of 145 µL min⁻¹ for 5 min so that analysed volume for each sample was 725 µL. This information will be added to section 2.3 in the revised manuscript.

L. 138. "Thus the increase, from the surface to the base of the euphotic layer, in phytoplankton biomass was ca. 2-fold, compared with ca. 8-fold for TChl a." Please, consider recalculating the biomass taking into account changes of biovolume with depth.

As shown in the response above, when phytoplankton biomass at the DCM is recalculated taking into account depth-related changes in SSC the new biomass data are virtually identical to the original ones. This results from several factors: i) the vertical changes in cell volume of picoeukaryotes were minor, ii) the change of *Synechococcus* and that of nanoeukaryotes were substantial but showed opposite trends, thus counterbalancing each other, and iii) the biomass of all groups measured with flow cytometry represent, on average, ≤40% of total phytoplankton biomass at the DCM.

Diatoms at the DCM L. 264. "The fucoxanthin to total chlorophyll a ratio (Fuco: TChl a) consistently increased below the upper 40-50 m in all long stations." From the changes in this ratio it is deduced that diatom contribution increases with depth. Fucoxanthin is also present in haptophytes and pelagophytes, two main components of phytoplankton with 19'hex-fuco and 19'but-fuco as their main diagnostic pigments, respectively. To make sure the increase in fucoxanthin is due to diatoms I would recommend calculating the vertical distribution of fucoxanthin: (19'hex-fuco + 19'but-fuco). The increase of this ratio with depth would be a more convincing evidence of a differential increase in diatoms. The images obtained with the Imaging Flow CytoBot should help to confirm that diatoms dominated or were very abundant in the DCM layer.

Following the reviewer's advice, we have calculated the fucoxanthin:(19'hex-fuco+19'but-fuco) ratio and verified that it increases consistently with depth. In fact, the vertical distribution of the fucoxanthin:(19'hex-fuco+19'but-fuco) ratio is nearly identical to that of the fucoxanthin:chlorophyll a ratio, which supports our conclusion of increased diatom contribution at the DCM. The new pigment ratio will be added to the supplementary information in the revised version of the manuscript. We will also add mosaics of all cells imaged by the IFCB in surface and DCM samples from the three long stations. These

mosaics show that diatoms were abundant at the DCM of all three stations and virtually absent in surface samples.

L. 375. ": : :this trend was associated with a significant increase in the contribution of diatoms to total phytoplankton biomass, which reached at least 30 % in the DCM of all stations, and was particularly high (nearly 50 %) in the most stratified station, located in the Ionian Sea." Please, re-check your estimates of diatom contribution at the DCM. Although I agree that diatoms can increase at the DCM, these values appear very high. In addition, the data of Crombet et al 2011 (cited in ms) show a patchy distribution of the Deep Silica Maximum and diatoms in the DCM of the Mediterranean.

It has to be noted that our cruise took place during late spring whereas the survey reported by Crombet et al. (2011) was conducted in summer, more than 1.5 months later in the year. We estimated the diatom contribution to total chl a by using three different pigment coefficients. The lowest pigment coefficient used (1.41, taken from Uitz et al. 2006), which gives a lower-bound estimate of diatom contribution, is derived from a large database covering a broad range of trophic situations and including the entire water column, not just the surface layer as is typically the case. We therefore consider that the resulting estimate of diatom contribution is robust. Note that station ION, which has the highest estimated contribution of diatoms in the DCM, is also the one that shows the highest abundance of diatoms in the IFCB images.

Primary production (PP) at the DCM L. 309. "In contrast, during PEACETIME the mean depths of the primary production maximum and the DCM coincided and only on 3 profiles was the primary production peak located above the DCM." The subsequent discussion does not present potential mechanisms to explain the discrepancy in PP estimates at the DCM between this and previous studies cited in the ms, which show a PP maximum above the DCM most of the time. It does argue that the high primary production at the DCM during PEACETIME was due not only to enhanced levels of phytoplankton biomass but also to the presence of a diatom-rich community characterized by high photosynthetic efficiency. It is a bit surprising that the same response has not been found in previous studies in the area. Could it be possible that the presence of diatoms with high photosynthetic efficiency at the DCM discussed by the authors is a consequence of the previous spring bloom at the surface and not a regular feature of the DCM in the Mediterranean? Estrada et al (1993. Variability of deep chlorophyll maximum characteristics in the Northwestern Mediterranean. Mar. Ecol. Prog. Ser. 92: 289–300) reported the occasional presence of diatoms from a decaying bloom that contributed significantly to the DCM biomass but with a very low photosynthetic efficiency, which seems typical of sinking cells. It seems that a large contribution of diatoms in the DCM layer is not a general feature of the Mediterranean Sea, and perhaps could explain the discrepancies in PP estimates at this depth with other studies.

We agree with the reviewer that the high diatom abundance and productivity observed during our cruise are not necessarily persistent features of the DCM in the Mediterranean sea. In fact, we end the Conclusions section by pointing out that future, high-resolution studies are needed to ascertain if the observed peak in productivity is a persistent feature of the DCM in the Mediterranean Sea. It may well be the case that, as indicated by the reviewer, the significant biomass contribution of diatoms observed at the DCM results from the sedimentation of the earlier spring bloom. In the revised version of the manuscript, we will re-write the relevant passages of sections 4.3 and 4.5 to point out that 1) our results have a limited temporal coverage and therefore cannot be used to ascertain if the deep productivity maxima are persistent during the stratification season and 2) it is possible that the enhanced biomass contribution of diatoms at the DCM results from the sedimentation of the spring bloom in the weeks prior to the cruise.

L. 340. "In contrast, during our survey the contribution of increased phytoplankton biomass was similar in all stations, including the one located in the Ionian Sea." An important conclusion is that DCM is a maximum of biomass and production in the Mediterranean, at least during the period of the study.

However, in 3 of the 4 profiles obtained in the Tyrrhenian Sea the biomass maximum is well above the DCM. This result is mentioned (line 235) but ignored throughout the ms. Moreover, it is difficult to explain how the PP maximum can be found at 70-80m, at the DCM and below the biomass maxima in these stations without a significant increase of nutrient supply. The correction that has been applied to short-term temperature variations to estimate PP at in-situ temperature from incubations at higher temperatures (about 5 C) could be discussed further to see if they may have distorted the results of the deep layers.

In two of the TYRR profiles mentioned (sampled on 18 and 19 May) there is indeed a disconnection between the deep PP maximum (located at the DCM depth) and the biomass maximum (located at 40 m) but it has to be noted that the magnitude of the deep PP peaks is minor. The rates of primary production measured on those 2 profiles at the DCM are only slightly higher than those measured at the surface (2 vs 1.7 and 3 vs 2.3 mgC m$^{-3}$ d$^{-1}$, respectively). On 17 May, PP at the DCM was twice as large as that at the surface, but biomass was also higher by a factor of 2. Large discrepancies between biomass and primary production would have resulted in anomalous values of biomass turnover rates, which were not found. The temperature correction used assumes a strong sensitivity of photosynthesis to temperature (Ea = 0.61 eV, approximately equivalent to $Q_{10}$ = 2.3), which only moderates the magnitude of the deep PP peaks, as explained in the first version of the manuscript (section 4.1).

L. 444. "Thus the surface BP (bacterial production) peak observed under in situ conditions was not due to dependence of organic carbon substrates but may have resulted in part from new N and P availability through dry atmospheric deposition." This explanation can be applied to phytoplankton as well. If atmospheric input of inorganic nutrients and recycling are the main reasons for vertical patterns of bacterial production, the same pattern should have been found for primary production (which is the pattern usually found by other studies in the Mediterranean cited in the ms).

The response to atmospheric deposition may not be necessarily symmetrical between phytoplankton and heterotrophic bacteria, as the latter tend to respond faster and more intensely to the nutrients injected from the atmosphere (see review of dust addition bioassays in Guieu et al. 2014). In fact, the superior ability of heterotrophic bacteria to compete for inorganic nutrients is also shown by the budget analysis and experimental observations of Van Wambeke et al. (2020), who concluded that dry atmospheric deposition could supply nearly 40% of the hetetrophic bacteria N demand in the upper mixed layer.

L. 335. "Therefore low nutrient availability, which is widespread in the global ocean (Moore et al., 2013), results not only in low phytoplankton biomass but also in slow growth rates." This conclusion is controversial in the scientific community. Another line of research with direct estimates of growth rates using mainly dilution experiments argue that, even with low nutrient concentrations, fast supply of nutrients from recycling results in the predominance of phytoplankton, usually of small size, with relatively high growth rates (Laws, E. A., 2013. Evaluation of in situ phytoplankton growth rates: A synthesis of data from varied approaches. Ann. Rev. Mar. Sci. 5: 247–268, and ref therein), although lower than those of taxa typical for bloom situations. Different optimal growth rates can be a function of taxonomical affiliation or size, among other reasons.

Other results from dilution experiments, not cited by Laws (2013), show that slow growth rates prevail in low-productivity waters. For instance, Landry et al. (2008) measured rates around 0.3 d$^{-1}$ in oligotrophic waters off Hawaii not affected by a cyclonic eddy, while finding rates as high as 0.6 d$^{-1}$ in stations inside the eddy. In another study, Landry et al. (2009) found euphotic layer-integrated phytoplankton growth rates of 0.1-0.2 d$^{-1}$ in oceanic, well-stratified stations off southern California, compared with rates of 0.2-0.5 d$^{-1}$ in stations within the coastal upwelling region (their Fig. 3). More recently, Armengol et al. (2019) obtained (also with the dilution method) mean growth rates around 0.3 d$^{-1}$ across the oligotrophic tropical Atlantic (10°N-0°S).

We agree with the reviewer that different taxa have different maximum growth rates. However, even though some strains of *Prochlorococcus* and *Synechococcus* may have relatively low maximum growth rates (<0.5 $d^{-1}$), picoeukaryotes of wide distribution such as *Ostreococcus* sp. and *Micromonas* sp. can indeed grow at rates ≥ 0.5 $d^{-1}$ (Six et al. 2008, Demory et al. 2019). The results of Berthelot et al. (2018) that we cite are especially relevant because they were based on measurements of isotope uptake by intact, single cells, thus avoiding some of the uncertainties involved in bulk methods. They found that growth rates of picoeukaryotes were 0.15-0.26 $d^{-1}$ in the North Pacific subtropical gyre compared with 0.42-0.50 $d^{-1}$ in stations within the California coastal current. Also, in situ experiments in HNLC waters have shown unequivocal  increases in growth rates once Fe limitation was removed (Boyd et al. 2008). Finally, flow cytometry measurements of single-cell fluorescence (a proxy for abundance of photosynthetic units) in subtropical gyres (Davey et al. 2008, Browning et al. 2017) show that investment in photosynthetic machinery increases markedly after nutrient addition, again supporting the view that nutrient limitation in oligotrophic regions causes physiological impairment and thus reduced growth rate.

Keep the same y-scale for fig 3g, h and i.

The scale in Fig 3i will be changed accordingly.

**References (not cited in the ms)**

Armengol et al. (2019) Planktonic food web structure and trophic transfer efficiency along a productivity gradient in the tropical and subtropical Atlantic Ocean. Sci Rep 9, Article No 2044

Boyd et al. (2008) Mesoscale Iron Enrichment Experiments 1993-2005: Synthesis and Future Directions Science 315 :612-617

Browning et al. (2017) Nutrient co-limitation at the boundary of an oceanic gyre. Nature, doi:10.1038/nature24063.

Cáceres et al. (2013) Phytoplankton Growth and Microzooplankton Grazing in the Subtropical Northeast Atlantic. PlosOne 8(7) e69159.

Davey et al. (2008) Nutrient limitation of picophytoplankton photosynthesis and growth in the tropical North Atlantic. Limnol Oceanogr 53:1722-1733.

Demory et al. (2019) Picoeukaryotes of the Micromonas genus: sentinels of a warming ocean. The ISME Journal, 13:132–146

Edwards et al. (2016) Phytoplankton growth and the interaction of light and temperature: A synthesis at the species and community level. Limnol Oceanogr, doi: 10.1002/lno.10282

Landry et al. (2008) Depth-stratified phytoplankton dynamics in Cyclone Opal, a subtropical mesoscale eddy. Deep Sea Res 55:1348-1359

Landry et al. (2009) Lagrangian studies of phytoplankton growth and grazing relationships in a coastal upwelling ecosystem off Southern California. Prog Oceanogr 83:208-216

Six et al. (2008) Contrasting photoacclimation costs in ecotypes of the marine eukaryotic picoplankter Ostreococcus. Limnol Oceanogr 53:255

Van Wambeke et al. (2020) Influence of atmospheric deposition on biogeochemical cycles in an oligotrophic ocean system. Biogeosciences, under revision.

---

## Author Comment (AC3) · 9 Nov 2020

**Comments by Anonymous Referee #3 and our responses**

This MS addresses the ubiquitous subsurface feature, the deep chlorophyll maximum, phytoplankton biomass and production, and heterotrophic prokaryotic production in the Mediterranean sea's stratified water column during the later spring season (May 10, 2017- June 11, 2017). This subsurface feature in the world ocean is known for long, more prominent in waters of lower latitude, are often found at nutracline depth well below the remote sensing reach, thus supports the importance of seaboard measurements to capture this feature. Chlorophyll a, an indicator of the phytoplankton biomass, is regulated by light, nutrient, etc. Here, the authors mainly aim to quantify photoacclimation's relative role and enhanced growth as an essential DCM mechanism. Secondly, the trophic coupling between phytoplankton and heterotrophic prokaryotic production is also addressed. Based on shipboard measurements in the Mediterranean sea, authors conclude that the DCM located at subsurface depth coincides with both biomass and primary production but not in growth rate and explains that the photoacclimation process leads to the increased chlorophyll a at the DCM. This study contributes vital insight into likely future ocean changes under the ocean warming scenario, thus merits publication of this work. However, I do not recommend a journal publishing this work in the present form. A few concerns about the methodology and the data interpretation need to be taken care of before considering this work for publication (see below).

We are grateful to this reviewer for their time and helpful comments.

Flow cytometry tool followed to obtain estimates of the carbon biomass in different size categories does not seem to have taken account of the autotrophic cells >150 microns in size neither their contribution is quantified, if minimal. The authors could have easily viewed these samples (>150micron) under the microscope to support the finding. If this were a significant observed, increased carbon biomass from the surface to the euphotic layer base would have been different and could lead to a different conclusion. The authors need to take care of this part in the section result and subsequently draw a conclusion at the end of the discussion section.

Although a 150-µm mesh is used to pre-filter the samples, particles with a length >150 µm can still be imaged by the IFCB. These include elongated, single cells (such as *Rhizosolenia* sp.) and diatom chains, both of which can be seen in the mosaics that will be included in the revised version of the manuscript. Size-abundance spectra obtained with microscopy image analysis in oligotrophic waters indicate that cells with a volume of $\geq 10,000$ µm$^3$ (assuming a cylindrical, elongated shape such as that of *Rhizosolenia*, this volume corresponds roughly to cells with a length of 150 µm and a diameter of 10 µm) contribute on average approximately 1% of total biovolume (Huete-Ortega et al. 2011, Marañón 2015). It is thus unlikely that the IFCB has significantly underestimated the total community biovolume. This point will be clarified in the revised version of the manuscript.

Also, it is unclear whether definite size beads were run on flow cytometry to conclude the mean cell diameter used for carbon calculation. It is essential to show the reader the error introduced by assuming the mean cell diameter (2um or 4 um or 6um).

As explained in the Methods section, C biomass estimates for small phytoplankton (measured with flow cytometry) assumed constant C content for each group of cells. Although estimates of cell biovolume based on the side scattering (SSC) signal were not routinely available for the cruise, we have examined a few profiles of SSC per cell at the long stations to assess depth-related changes in cell biovolume of *Synechococcus*, picoeukaryotes and nanoeukaryotes. As shown in the plots and table below, we found that cell biovolume of nano- and pico-eukaryotes decreased with depth whereas the opposite was true for *Synechococcus*.

[Figure]

Mean (and standard deviation) of the side-scatering signal per cell of different groups in the surface layer (0-40 m) and at the DCM (including also the sample obtained immediately above the DCM) in the three long stations (data from all stations were pooled together).

|  | SSC per cell | | |
|---|---|---|---|
|  | Nanoeukaryotes | Picoeukaryotes | *Synechococcus* |
| Surface (n = 13) | 1.06 (0.11) | 0.24 (0.04) | 0.052 (0.010) |
| DCM (n = 11) | 0.64 (0.15) | 0.21 (0.03) | 0.074 (0.017) |

We have recalculated the total biomass of phytoplanton at the DCM taking into account these depth-related changes in cell volume of pico and small nanophytoplankton. Specifically, the value of C biomass per cell used in the original calculations was multiplied by the observed DCM to surface SSC ratio for each group, which was 0.61 for nanoeukaryotes, 0.87 for picoeukaryotes and 1.42 for *Synechococcus*. The figure below shows that taking into account these changes in cell volume with depth has negligible effects on the estimated total phytoplankton biomass at the DCM:

[Figure]

On the other hand, while calculating Fucoxanthin to total chlorophyll a ratio calculation, I find authors have ignored that besides diatoms, Phaeocystis spp. are also potential sources of Fucoxanthin (see Latasa and Bidigare, 1998) instead accounted to diatom community.

To examine the possibility that the increase in fucoxanthin with depth may have reflected an increased abundance in other fucoxanthin-containing groups such as haptophytes and pelagophytes, we have calculated the fucoxanthin:(19'hex-fuco+19'but-fuco) ratio. We found that the vertical distribution of the fucoxanthin:(19'hex-fuco+19'but-fuco) ratio is nearly identical to that of the fucoxanthin:chlorophyll a ratio, which supports our conclusion of increased diatom contribution at the DCM. The new pigment ratio will be added to the supplementary information in the revised version of the manuscript. We will also add mosaics of all cells imaged by the IFCB in surface and DCM samples from the three long stations. These mosaics show that diatoms were abundant at the DCM of all three stations and virtually absent in surface samples.

Furthermore, the presence of divinyl chl a, a marker for prochlorophyte, seems to have ignored and accounted for diatoms. Suggest authors revisit HPLC based pigment (depth-wise) analyses to rule out Prochlorococcus community is not missed out. In my opinion, low light-adapted Prochlorococcus at the DCM may be sizably contributing to the DCM community.

Following the reviewer's advice, we have assessed the potential contribution of *Prochlorococcus* to total phytoplankton biomass by examining previously unused data obtained with flow cytometry. Applying a cell C content of 0.06 pgC cell$^{-1}$ (based on Buitenhuis et al. 2012), the typical *Prochlorococcus* abundances measured at the DCM (ca. 40,000 cell mL$^{-1}$) represent a C biomass of ca. 2.4 mgC m$^{-3}$. By contrast, *Prochlorococcus* was undetected in the upper layers (0-30 m). In the revised version of the manuscript, total phytoplankton biomass will be recalculated taking into account also the contribution of *Prochlorococcus.*

**References**

Marañón (2015) Cell size as a key determinant of phytoplankton metabolism and community structure. Annual Review of Marine Science, 7, 241-264. doi: 10.1146/annurev-marine-010814-015955

Huete-Ortega et al. (2011) Isometric size-scaling of metabolic rate and the size abundance distribution of phytoplankton. Proceedings of the Royal Society B, 279, 1815-1823. doi:10.1098/rspb.2011.2257.